# HLH-11 modulates lipid metabolism in response to nutrient availability

Yi Li[1,2,3], Wanqiu Ding[4], Chuan-Yun Li [4] & Ying Liu [1,2,5✉]

The ability of organisms to sense nutrient availability and tailor their metabolic states to withstand nutrient deficiency is critical for survival. To identify previously unknown regulators that couple nutrient deficiency to body fat utilization, we performed a cherry-picked RNAi screen in *C. elegans* and found that the transcription factor HLH-11 regulates lipid metabolism in response to food availability. In well-fed worms, HLH-11 suppresses transcription of lipid catabolism genes. Upon fasting, the HLH-11 protein level is reduced through lysosome- and proteasome-mediated degradation, thus alleviating the inhibitory effect of HLH-11, activating the transcription of lipid catabolism genes, and utilizing fat. Additionally, lipid profiling revealed that reduction in the HLH-11 protein level remodels the lipid landscape in *C. elegans*. Moreover, TFAP4, the mammalian homolog of HLH-11, plays an evolutionarily conserved role in regulating lipid metabolism in response to starvation. Thus, TFAP4 may represent a potential therapeutic target for lipid storage disorders.

[1] State Key Laboratory of Membrane Biology, Institute of Molecular Medicine, Peking University, 100871 Beijing, China. [2] Peking-Tsinghua Center for Life Sciences, Peking University, 100871 Beijing, China. [3] Academy for Advanced Interdisciplinary Studies, Peking University, 100871 Beijing, China. [4] Laboratory of Bioinformatics and Genomic Medicine, Institute of Molecular Medicine, Peking University, 100871 Beijing, China. [5] Beijing Advanced Innovation Center for Genomics, Peking University, 100871 Beijing, China. ✉email: ying.liu@pku.edu.cn

n their natural environments, organisms are actively challenged with fluctuating food supplies. Their ability to sense nutrient availability and adjust their metabolic programs to adapt to complex nutritional conditions is critical for survival. When they are well fed, organisms utilize external nutrients, while in periods of starvation, they can sense nutrient deficiency and activate coherent transcriptional responses to express catabolic enzymes. This results in metabolic reprogramming so that internal energy reserves are used to survive starvation. These nutrient-based transcriptional responses, which ensure metabolic tuning, are conserved across phyla[1–10].

A major energy reserve of most organisms is their body fat. Therefore, active and rapid consumption of fat stores is an ancient and conserved mechanism for surviving food deficiency[11], which results in decreased body fat levels[7–10]. Fat loss in starved animals is a result of imbalanced fat degradation and synthesis. The initial phase of fat degradation starts with lipophagy or lipolysis, which removes fatty acids from triacylglycerols (TAGs)[12,13]. Free fatty acids are then cyclically shortened via fatty acid oxidation to produce acetyl-CoA. Conversely, synthesis of lipids starts from the synthesis of fatty acids from acetyl-CoA. Fatty acids are further elongated or desaturated by elongases or desaturases and subsequently incorporated into TAGs.

Due to high conservation of metabolic networks and nutrient-sensing mechanisms across phyla, *Caenorhabditis elegans* (*C. elegans*) represents an ideal model organism to dissect the mechanisms that sense nutrient deficiency and maintain energy balance through the regulation of fat metabolism. In response to fasting, *C. elegans* activates the expression of lipid catabolic enzymes, including ATGL-1 in lipolysis[14], LIPLs in lipophagy[15] and ACS-2 in β-oxidation, and downregulates the expression of FAT-7 in fatty acid biosynthesis[4]. Several regulators have been identified in *C. elegans* that link transcriptional changes to energy or nutrient demands. For example, nuclear hormone receptor NHR-49, an ortholog of peroxisome proliferator-activated receptor-α (PPARα)[16–19], is required for the regulation of "fasting response genes," including those involved in lipid metabolism[4]. Moreover, the basic helix–loop–helix (bHLH) family transcription factor HLH-30, which is homologous to mammalian transcription factor EB (TFEB)[3], has been show to activate transcription of lipid catabolism genes and promote fat utilization upon food withdrawal[15]. In contrast to NHR-49 and HLH-30, which act positively to promote fat breakdown under nutrient-depleted conditions, the Mondo family transcription factor MXL-3 antagonizes the activity of HLH-30 to repress lysosomal lipolysis when food is sufficient[15]. In addition, the transcription factor SBP-1, which is the *C. elegans* homolog of sterol regulatory element binding protein[20], promotes lipid synthesis and facilitates fat storage when worms are exposed to sugar-rich media[21]. The mechanisms that couple nutrient availability to fat utilization demand extensive study and full characterization.

To further investigate the mechanisms that underlie the starvation response and fat metabolism, we generated two fluorescent reporter strains *Plipl-3::gfp* and *Pacs-2::gfp* in *C. elegans* and carried out RNA interference (RNAi) screening to identify gene inactivations that affect green fluorescent protein (GFP) expression. We found that deficiency of HLH-11, a bHLH transcription factor, activated GFP expression in reporter strains even under ad libitum feeding conditions. Transcriptome analysis of wild-type, *hlh-11* knockout (KO), and *hlh-11* overexpression (OE) animals revealed that, when food is available, HLH-11 acts to suppress the transcription of lipid catabolism genes. However, during starvation, HLH-11 protein is degraded, resulting in active transcription of lipid catabolism genes and reprogramming of the lipid landscape in *C. elegans*. More importantly, we showed that TFAP4 (transcription factor AP-4), a human ortholog of HLH-11, plays

an evolutionarily conserved role in coupling nutrient availability with fat metabolism. Excess weight and obesity are major risk factors for a number of chronic diseases, including diabetes and cardiovascular diseases[22,23]. The discovery of TFAP4/HLH-11 may lead to medical interventions that mimic the effects of dietary restriction. TFAP4 is a potential therapeutic target for preventing lipid metabolism diseases.

## Results

**Deficiency of *hlh-11* mimics a starvation response.** Successful survival of organisms requires timely and accurate sensing of nutrient levels and reprogramming of metabolism in response to nutrient deficiency. Like higher eukaryotes, *C. elegans* responds to starvation by activating lipid utilization. This results in the reduction of body fat storage as revealed by Oil Red O (ORO), a lipophilic dye that stains fat droplets in all major fat storage organs of worms (Fig. 1a). Consistent with the decreased body fat stores under starved conditions, fasted animals had significantly increased transcript levels of *lipl-3*, encoding a lysosomal lipase that breaks down lipid-droplet fats to fatty acids, and *acs-2*, encoding an acyl-CoA synthetase that catalyzes the activation of fatty acids to fatty acyl-CoAs for β-oxidation[4,6,15] (Fig. 1b, c). Because the degradation of lipids, in response to fasting, consists of two steps, lipophagy/lipolysis and β-oxidation, we generated two starvation-responsive reporter strains in *C. elegans*, *Plipl-3::gfp* and *Pacs-2::gfp*. We reasoned that, by using both of the reporters, our screen will uncover previously unknown regulators that function in upstream signaling to modulate starvation-induced lipid catabolism. Consistent with the endogenous expression levels, GFP signals from these two reporter strains were dramatically elevated upon food deprivation (Fig. 1d, e).

We carried out a cherry-picked RNAi screen to examine changes in GFP expression and thus identify gene inactivations that activate the *lipl-3* and *acs-2* reporters under ad libitum feeding conditions. The cherry-picked RNAi library consists of 2252 genes (Supplementary Data 1), including kinases, phosphatases, transcriptional factors, and nuclear hormone receptors. These are likely candidates for transduction of cellular signals (the kinases and phosphatases) and activation of target genes (the transcription factors and nuclear hormone receptors). In addition, the cherry-picked library contains genes that are suggested by previous reports to play a role in feeding responses[24], regulation of body fat[25], and lysosomal function[26]. Interestingly, we found that knockdown of 57 genes, including those encoding mitochondrial ribosomal proteins, tRNA ligases, and transcription factors, mimicked the starvation response and activated the GFP reporters in ad libitum-fed worms (Supplementary Data 2). Among these hits, we first focused on a transcription factor named HLH-11 for three reasons. First, RNAi knockdown of *hlh-11* caused relatively high induction of the *lipl-3* and *acs-2* reporters (Supplementary Fig. 1a, b). *hlh-11* RNAi efficiency was verified via both imaging and immunoblotting (Supplementary Fig. 1c, d). Second, *hlh-11* encodes a bHLH transcription factor[27,28], which is likely to modulate a large set of downstream genes and play an important role in reprogramming lipid metabolism in response to nutrient deficiency. Third, the *cis*-regulatory motif of HLH-11 has been found in the promoter of *atgl-1*, a triglyceride lipase that catalyzes lipolysis in response to fasting[27].

To rule out the possibility that RNAi may have off-target effects on other genes, we employed a CRISPR/Cas9 approach to generate an *hlh-11* knockout (KO) strain (Supplementary Fig. 1e). The *hlh-11* gene has multiple transcripts, so we targeted a protein coding region that is shared by them all. In the KO animals, 37 bp

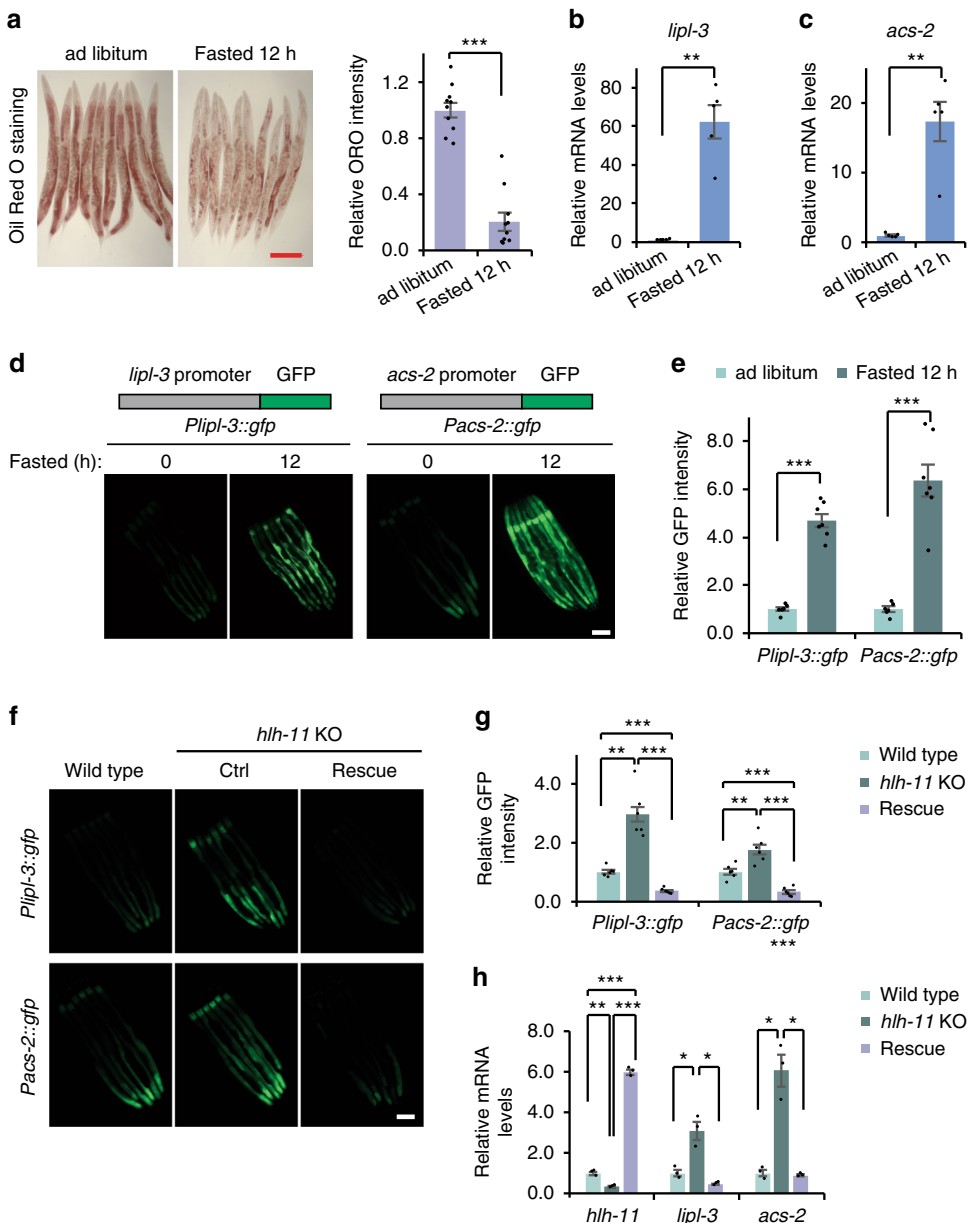

**Fig. 1 HLH-11 regulates the transcription of starvation response genes. a** Oil Red O staining and quantification of wild-type worms (day 1 adult) fed *ad libitum* or fasted for 12 h. $n = 10$ worms examined per condition. Data shown here are from one experiment. Four independent experiments were performed with similar results. Scale bar, 200 μm. ***$p = 2.7E{-}08$. **b, c** qPCR analyses of endogenous mRNA levels of *lipl-3* (**b**) or *acs-2* (**c**) in wild-type worms fed ad libitum or fasted for 12 h. **$p = 0.0020$ (**b**), **$p = 0.0043$ (**c**). $n = 5$ biologically independent samples. **d** Fluorescence images of *Plipl-3::gfp* and *Pacs-2::gfp* worms fed ad libitum or fasted for 12 h. Four independent experiments were performed with similar results. Scale bar, 100 μm. **e** Quantification of GFP intensity in **d**. $n = 6$–7 worms examined. ***$p = 3.6E{-}6$ (left), 0.00018 (right). **f** Fluorescence images of *Plipl-3::gfp* and *Pacs-2::gfp* reporters in wild-type or *hlh-11* knockout (KO) worms or *hlh-11* knockout worms with exogenous expression of HLH-11 (rescue). Four independent experiments were performed with similar results. Scale bar, 100 μm. **g** Quantification of GFP intensity in **f**. $n = 6$ worms examined. The exact $p$ values are provided in the Source data. **h** qPCR analyses of endogenous mRNA levels of *hlh-11, lipl-3* or *acs-2* in wild-type, *hlh-11* KO, or *hlh-11* rescue worms. $n = 3$ biologically independent samples. The exact $p$ values are provided in the Source data. Error bars indicate mean ± SEM. Statistical analyses were performed by Student's *t* test (unpaired, two tailed), *$p < 0.05$; **$p < 0.01$; ***$p < 0.001$. Source data are provided as a Source data file.

of DNA in the second exon of *hlh-11* was replaced by 17 bp (Supplementary Fig. 1e), resulting in a frameshift that abolishes the function of HLH-11. Consistent with the RNAi results, GFP signals were elevated in *hlh-11* KO animals expressing the *Plipl-3::gfp* and *Pacs-2::gfp* reporters (Fig. 1f, g). In addition, we generated an extrachromosomal strain that expresses wild-type *hlh-11* under its own promoter and introduced it into *hlh-11* KO animals. Complementing HLH-11 function was sufficient to suppress GFP reporter induction in *hlh-11* KO worms (Fig. 1f, g).

To further validate that deficiency of *hlh-11* truly affects endogenous *lipl-3* and *acs-2* expression, we used reverse transcription quantitative polymerase chain reaction (RT-qPCR) to examine *lipl-3* and *acs-2* transcript levels in wild-type animals and *hlh-11* KOs. *hlh-11* KO animals showed an increase in *lipl-3* and *acs-2* mRNA levels, and *hlh-11* rescue was able to suppress the induction of *lipl-3* and *acs-2* (Fig. 1h), confirming the results obtained with the fluorescent reporter strains. We further tested whether HLH-11, as a transcriptional repressor, directly binds to

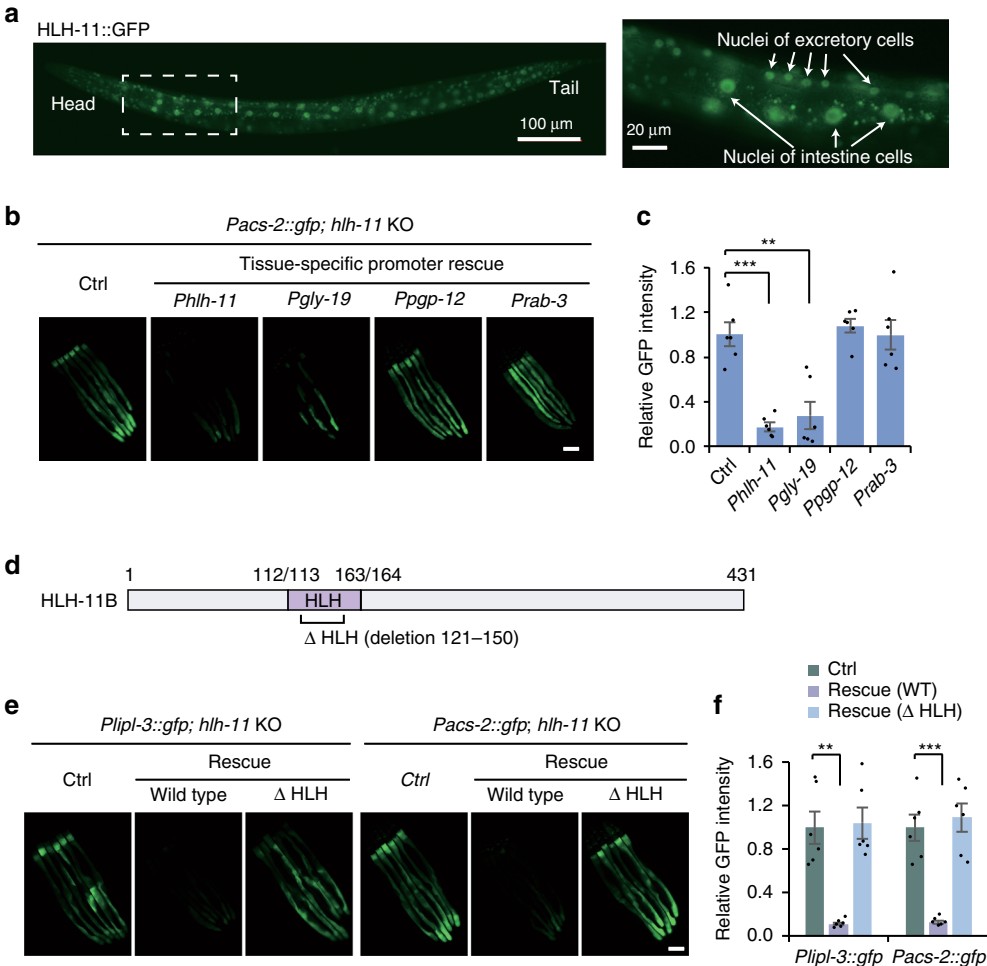

**Fig. 2 HLH-11 locates in the nuclei and functions cell autonomously in the intestine to regulate *lipl-3* and *acs-2* expression. a** Fluorescence images of *Phlh-11::HLH-11::TY1::EGFP::3xFLAG* worms. Eight worms were examined with similar results. **b** Fluorescence images of the *Pacs-2::gfp* reporter in *hlh-11* KO worms with exogenous HLH-11 expression driven by the promoter of *hlh-11*, *gly-19* (intestine), *pgp-12* (excretory cell), or *rab-3* (pan-neuronal). Three independent experiments were performed with similar results. Scale bar, 100 μm. **c** Quantification of GFP intensity in **b**. $n = 6$ worms examined per condition. ***$p = 0.0003$; **$p = 0.0013$. **d** A schematic diagram depicting the HLH domain in HLH-11B. **e** Fluorescence images of the *Plipl-3::gfp* or *Pacs-2::gfp* reporter in *hlh-11* KO animals with expression of wild-type HLH-11 or mutant HLH-11 with the HLH domain deleted (Δ HLH). Three independent experiments were performed with similar results. Scale bar, 100 μm. **f** Quantification of GFP intensity in **e**. $n = 6$ worms examined per condition. **$p = 0.0017$; ***$p = 0.0007$. Error bars indicate mean ± SEM. Statistical analyses were performed by Student's *t* test (unpaired, two tailed), **$p < 0.01$; ***$p < 0.001$. Source data are provided as a Source data file.

the promoters of *lipl-3* and *acs-2* to repress their expression. Indeed, chromatin immunoprecipitation (ChIP)-qPCR experiments indicated that HLH-11 can directly bind to the promoters of *lipl-3* and *acs-2* (Supplementary Fig. 1f). Taken together, these results suggest that deficiency of HLH-11 mimics a starvation response to induce the expression of *lipl-3* and *acs-2*.

**HLH-11 functions in the nucleus of intestine cells**. Three protein isoforms of HLH-11 have been reported (wormbase). We obtained an *hlh-11* deletion allele *ok2944*, which affects isoforms a/b but not isoform d (Supplementary Fig. 2a) and introduced this mutant allele into the *Plipl-3::gfp* or *Pacs-2::gfp* reporter strains. The mutant allele was capable of elevating the GFP signals in *Plipl-3::gfp* and *Pacs-2::gfp* to a similar extent as the KO strain, which lacks all three isoforms (Supplementary Fig. 2b, c). This suggests that *hlh-11* isoforms a/b might be responsible for the transcriptional regulation of *lipl-3* and *acs-2*.

In order to characterize the subcellular localization and body distribution of HLH-11, we obtained a strain expressing HLH-11 translationally fused with GFP. Expression of *Phlh-11::HLH-11::*

GFP was observed in the nuclei of intestinal cells, neurons, and excretory cells (wormbase, Fig. 2a). These cells can perform endocrine functions to coordinate *C. elegans* metabolism in response to nutrient levels. To test in which tissue HLH-11 functions to modulate the expression of lipid catabolism genes (e.g., *acs-2*), we expressed HLH-11 under intestine-, excretory cell-, and neuron-specific promoters (*gly-19*, *pgp-12*, and *rab-3*, respectively; Supplementary Fig. 2d). Tissue-specific expression of HLH-11 in the intestine, rather than in neuronal or excretory cells, was sufficient to suppress activation of *Pacs-2::gfp* in *hlh-11* KO animals (Fig. 2b, c).

HLH-11 belongs to a family of bHLH transcription factors[27,28]. The signature HLH domain normally mediates protein dimerization and is required for DNA binding. Deletion of the HLH domain in HLH-11 did not affect its nuclear localization (Supplementary Fig. 2e), but the mutant protein failed to suppress the induction of the *Plipl-3::gfp* or *Pacs-2::gfp* reporter in *hlh-11* KO animals (Fig. 2e, f). This suggests that the HLH domain is indispensable for transcriptional regulation of *lipl-3* and *acs-2*.

**HLH-11 negatively regulates the expression of lipid catabolism genes.** Coordinated regulation of enzymes within the same metabolic pathways is required in order to achieve an appropriate response to different nutritional conditions and metabolic demands. In *C. elegans*, the intestine is a major metabolic organ essential for both lipid synthesis and utilization. To avoid futile cycles of lipid synthesis and breakdown within the same tissue, a mechanism must have evolved to precisely control lipid anabolic and catabolic processes.

To gain further insights into whether lipid metabolism genes are coordinately regulated by HLH-11, we performed mRNA-seq analyses to compare the transcriptomes of wild-type, *hlh-11* KO, and *hlh-11* OE worms. A large set of genes were differentially expressed in *hlh-11* KO worms compared with wild-type animals (2637 genes upregulated and 2845 genes downregulated, Supplementary Data 3). Heat maps showed that *hlh-11* OE animals were more similar to wild type: the number of differentially expressed genes in *hlh-11* OE worms compared with wild-type animals was relatively small (107 genes upregulated, 237 genes downregulated; Fig. 3a and Supplementary Data 3).

Gene ontology analysis revealed that genes involved in lipid metabolism indeed appeared to be upregulated in *hlh-11* KO worms and downregulated in *hlh-11* OE worms (Fig. 3b, c). We systematically compared the expression of the lipid metabolism genes in *hlh-11* KO vs wild-type animals, and *hlh-11* OE vs wild-type animals. Interestingly, we found that, in addition to *lipl-3*, several other lipase genes, especially those that have been shown to be induced during fasting[15], such as *lipl-1,2,4,6* were also induced in *hlh-11* KO worms (Fig. 3d). Conversely, the lipase genes *lipl-1,2,7* and *lipl-3* were suppressed in *hlh-11* OE worms (Fig. 3d). During lipid catabolism, after breakdown of body fats by lysosomal lipases to generate free fatty acids (FFAs), the carbon–carbon bonds of FFAs are further broken down through β-oxidation in mitochondria and peroxisomes to provide energy for cellular activities. Two reactions are repetitively carried out in the β-oxidation process, which require the key enzymes acyl-CoA synthases (ACS) and acyl-CoA dehydrogenases (ACDH). The *C. elegans* genome contains around 20 ACS genes and >10 ACDH genes. Consistent with the observation that HLH-11 modulates lipase gene expression, downstream fatty acid β-oxidation pathway genes (e.g., ACSs and ACDHs) were also significantly induced in *hlh-11* KO worms (Fig. 3d). Taken together, these results indicate that HLH-11 negatively regulates fat catabolism genes.

We also analyzed the expression changes of fat synthesis genes in wild-type, *hlh-11* KO, and *hlh-11* OE animals, including those involved in converting acetyl-CoA to malonyl-CoA and eventually to C16 palmitate. Unlike fat catabolism, fat synthesis is only controlled by a handful of key enzymes, such as POD-2 (acetyl-CoA carboxylase) and FASN-1 (fatty acid synthase). C16 palmitate generated by FASN-1 is elongated to C18 stearic acid via the actions of elongases or converted to unsaturated fatty acids via the actions of desaturases (FATs). Unlike lipid catabolism genes, the transcript levels of fat synthesis genes such as *pod-2*, *fasn-1*, and the majority of elongases and desaturases remained unchanged in both *hlh-11* KO and *hlh-11* OE worms (Fig. 3d). Notably, *fat-7*, a desaturase coding gene that has been reported to be suppressed in fasted worms[4,6], was also suppressed in *hlh-11* KO worms (Fig. 3d).

Furthermore, we found that HLH-11 specifically regulates lipid metabolism genes. Expression levels of genes involved in other metabolic processes, such as glycolysis and the tricarboxylic acid (TCA) cycle, remained unchanged in *hlh-11* KO or OE animals (Supplementary Fig. 3a). Interestingly, we also noticed that genes involved in the innate immune response, including those

encoding C-type lectins, were also upregulated in *hlh-11* KO worms (Fig. 3b and Supplementary Fig. 3b). It has been shown in *C. elegans* that dietary restriction extends worm lifespan through modulation of the innate immune pathway[29]. It has also been shown that obesity, type 2 diabetes, or a western diet can reprogram the innate immune system and increase pathogenic inflammation[30,31], which suggests that immunity may be affected by metabolic status. Due to the potential importance of the cross-talk between innate immunity and metabolic state, it is interesting that we found a transcriptional regulator of lipid catabolism that also affects the expression of innate immune genes. Therefore, we took a closer look at the expression of genes in the canonical innate immune pathways of *C. elegans* (Supplementary Fig. 3b). We found that *zip-2* and its target gene *irg-1* (ref. [32]) and the *fshr-1* target gene *F01D5.5* (ref. [33]) were upregulated in *hlh-11* KO worms (Supplementary Fig. 3b). Conversely, transcript levels of *pmk-1* pathway genes such as *K08D8.5*, *C17H12.8*, *T24B8.5*, and *F55G11.2* (refs. [29,34]) were unchanged or downregulated in *hlh-11* KO worms (Supplementary Fig. 3b). It has been reported that 296 genes are upregulated when worms are treated with *Pseudomonas aeruginosa* PA14 (ref. [34]). Among these 296 genes, 104 genes were also upregulated in *hlh-11* KO worms (Supplementary Fig. 3c). In addition, a significant number (34.7%) of innate immune response genes elevated in *hlh-11* KO worms were induced by PA14 treatment (Supplementary Fig. 3d). It will be of particular interest in the future to directly test whether *hlh-11* plays a critical role in modulating innate immunity in response to different nutrient levels.

**HLH-11 controls organismal body fat levels.** Because HLH-11 negatively regulates the expression of many lipid catabolism genes, we reasoned that changes in HLH-11 levels should affect body fat accumulation. Indeed, a significant reduction in body fat stores, as detected by the intensity of ORO staining, was observed in *hlh-11* KO animals (Fig. 4a, b). Lysosomal fats, detected by Nile Red staining, were also decreased in *hlh-11* KOs (Supplementary Fig. 4a, b). Consistent with the observation that expression levels of most lipid metabolism genes were not affected in *hlh-11* OE animals, OE of HLH-11 did not significantly alter body fat levels (Fig. 4a, b and Supplementary Fig. 4a, b).

Besides serving as an energy supply, lipids can also be transferred to embryos in the form of yolk to support progeny development. It is possible that loss of lipid storage in *hlh-11* KO mutants is due to excess yolk production. To test this possibility, we first measured the fat content in *hlh-11* KO males or L4 stage worms, which do not transfer lipids to fulfill the energy demands of embryos. The fat loss phenotype was also observed in *hlh-11* KO males or L4 stage worms (Supplementary Fig. 4c–f). Moreover, based on the mRNA-seq data, we found that KO of *hlh-11* suppresses the transcription of all six Vitellogenin genes (Supplementary Fig. 4g). Therefore, vitellogenesis, the process of yolk formation, is unlikely to be activated under HLH-11 deficiency. Lastly, ORO staining revealed that the lipid levels were not elevated in embryos of *hlh-11* KO animals (Supplementary Fig. 4h). Taken together, these results suggest that *hlh-11* inactivation initiates fat breakdown to support the energetic demands of parental worms, resulting in fat loss from these animals.

We further performed biochemical extraction, followed by mass spectrometry, to evaluate the impact of HLH-11 on the composition and spectrum of lipids in *C. elegans*. Body fat in *C. elegans* is stored mainly in the intestine, and previous lipid profiling studies showed that this intestinal fat is predominantly in the form of TAGs[14]. Consistent with ORO staining, we found that *hlh-11* KO worms had significantly less TAGs (Fig. 4c and Supplementary Fig. 4i). We further examined changes of 231 individual lipids within the TAG categories. Approximately 40%

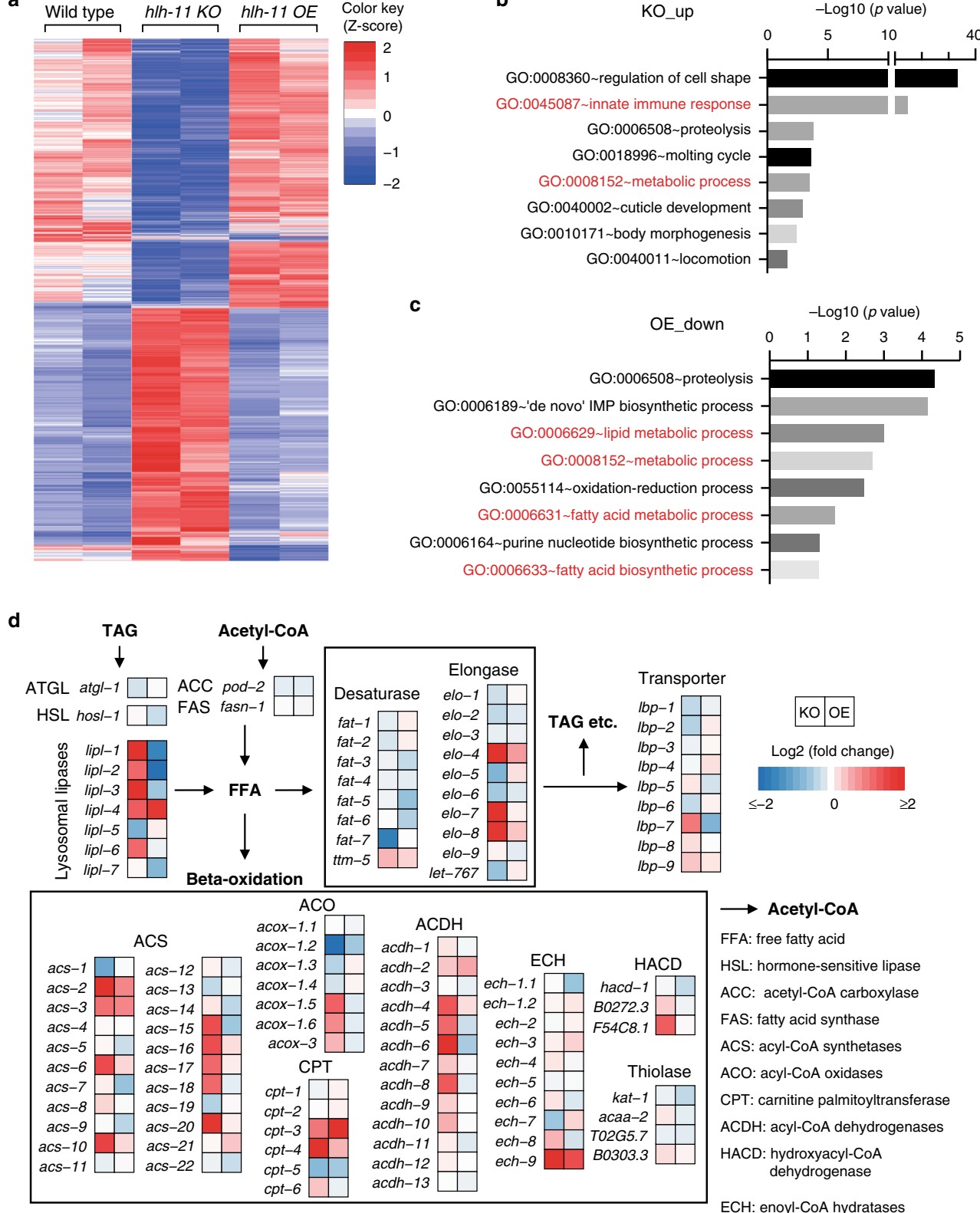

**Fig. 3 HLH-11 negatively regulates the transcription of lipid catabolism genes. a** Heat map of differentially expressed genes in *hlh-11* knockout (KO) or *hlh-11* overexpression (OE) worms compared to wild-type worms. Two replicates are shown for each genotype. *p* values were calculated by two-sided Wald test and corrected for multiple testing using the Benjamini–Hochberg method. Genes with an adjusted *p* value <0.05 were selected as differentially expressed genes. **b, c** Gene ontology (GO) enrichment analysis of the genes upregulated in *hlh-11* KO worms (**b**) or downregulated in *hlh-11* OE worms (**c**). *p* values were calculated by one-sided Fisher's exact test modified by the DAVID tool and corrected for multiple testing using the Bonferroni method. GO terms related to immune response and metabolic process are highlighted in red. **d** Heat maps of fold changes for mRNA levels of lipid metabolism genes in *hlh-11* KO and OE worms. The fold changes are calculated by dividing the mRNA level of each indicated gene in *hlh-11* KO or OE animals by that of wild-type animals. Source data are provided as a Source data file.

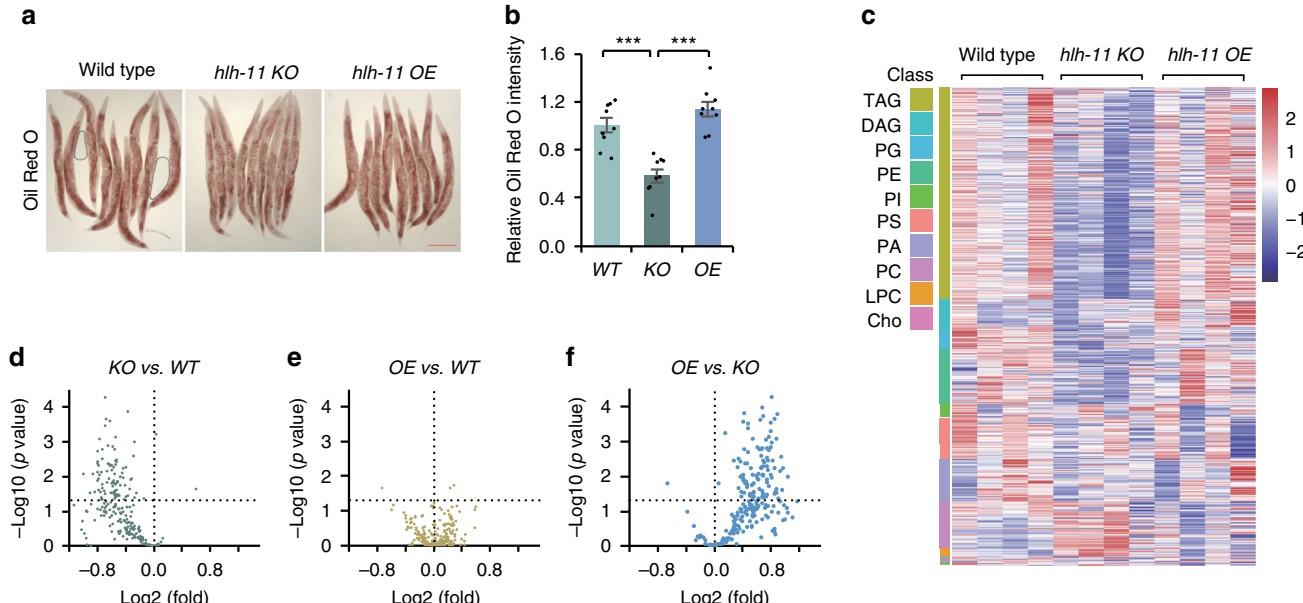

**Fig. 4 Knockout of HLH-11 affects lipid storage. a** Oil Red O staining of wild-type, *hlh-11* KO, and *hlh-11* OE worms (day 1 adult). Three independent experiments were performed with similar results. Scale bar, 200 μm. **b** Quantification of Oil Red O signals from the upper intestine in **a**. $n = 9$ worms examined per condition. Error bars indicate mean ± SEM. Statistical analyses were performed by Student's *t* test (unpaired, two tailed), ***$p = 9.5E-5$ (KO vs WT), ***$p = 3.0E-6$ (OE vs KO). **c** Heat map showing the levels of lipids detected by mass spectrometry in wild-type, *hlh-11* KO, and *hlh-11* OE worms. TAG triacylglycerols, DAG diacylglycerols, PG phosphatidylglycerols, PE phosphatidylethanolamines, PI phosphatidylinositols, PS phosphatidylserines, PA phosphatidic acids, PC phosphatidylcholines, LPC lyso-phosphatidylcholines, Cho free cholesterols. **d–f** Volcano plots showing fold changes of lipids in the TAG categories. **d** *hlh-11* KO compared with wild-type animals; **e** *hlh-11* OE compared with wild-type animals; **f** *hlh-11* OE compared with *hlh-11* KO. *p* values are calculated using one-sided Tukey's HSD test. Horizontal dashed line indicates $p = 0.05$. Source data are provided as a Source data file.

of TAGs were significantly reduced in *hlh-11* KO worms compared to wild-type animals (Fig. 4d), while levels of TAGs in *hlh-11* OE worms were comparable with wild-type (Fig. 4e). Moreover, ~48% of TAGs were significantly increased in *hlh-11* OE worms, compared to *hlh-11* KO worms (Fig. 4f). These results strongly suggest that HLH-11 affects lipid storage in *C. elegans*.

**Degradation of HLH-11 protein occurs in response to starvation.** Transcriptional responses linked to lipid synthesis or utilization are activated according to nutrient availability. Since HLH-11 negatively regulates lipid catabolism genes and remodels the lipid landscape in *C. elegans*, we wondered whether HLH-11 plays a role in nutrient sensing to initiate a fasting transcriptional program that utilizes fat reserves to survive starvation. A translational fusion of HLH-11 to EGFP revealed that, upon starvation, the HLH-11::GFP signal gradually decreased in the nuclei of intestinal cells and remained close to undetectable even after 24 h of fasting (Fig. 5a, b). In addition, protein levels of HLH-11 were also reduced in response to food deprivation, as revealed by immunoblotting (Fig. 5c).

Certain genetic perturbations mimic starvation to activate metabolic reprogramming. For example, *eat-2* mutants, which lack a ligand-gated ion channel required for normal pharyngeal muscle function, have reduced pharyngeal pumping and food intake and have been commonly used as a genetic diet-restriction mimetic[35]. Similar to food deprivation, we found that mutation of *eat-2* significantly reduced the HLH-11 protein level (Supplementary Fig. 5a, b). We also employed a *Phlh-11::GFP* transcriptional fusion strain to test whether transcription of *hlh-11* is also regulated by nutrient conditions. Unlike the translational fusion reporter, GFP signals driven by the *hlh-11*

promoter remained unchanged in response to food deprivation (Fig. 5d, e). Consistent with this result, endogenous transcript levels of *hlh-11* were not regulated in a starvation-responsive manner (Fig. 5f). Taken together, these results suggest that HLH-11 might be regulated translationally or posttranslationally in response to food deprivation.

To further discern the molecular mechanism that regulates HLH-11 following fasting, we tested whether HLH-11 is degraded by lysosomes or proteasomes in response to starvation. Worms treated with the lysosomal acidity neutralizer $NH_4Cl$, or carrying the *glo-4(ok623)* mutation that results in lack of functional intestinal lysosomes, still showed a reduced level of HLH-11 protein following fasting (Fig. 5g and Supplementary Fig. 5c). In addition, treating *C. elegans* with the proteasome inhibitor Bortezomib had no effect on HLH-11 protein levels before or after food deprivation (Supplementary Fig. 5c). Interestingly, treating worms with both $NH_4Cl$ and Bortezomib, or treating *glo-4(ok623)* mutants with Bortezomib, which impairs both lysosomes and proteasomes in *C. elegans*, was able to suppress the reduction of HLH-11 upon fasting (Fig. 5g and Supplementary Fig. 5c). This suggests that, under starvation conditions, both lysosomes and proteasomes are employed to link HLH-11 protein levels with nutrient availability.

mTOR (mechanistic target of rapamycin) and AMPK are key factors in two major nutrient-sensing pathways. To see whether any of the known nutrient-sensing pathways are involved in regulating HLH-11, we inactivated *let-363/Ce*.TOR by RNAi or activated AMPK through OE of AAK-2 (*C. elegans* APMK subunit). These two approaches have been used to mimic the effects of starvation[29,36]. We found that inhibition of *Ce*.TOR reduced the HLH-11 protein level (Supplementary Fig. 5d, e), whereas activation of AMPK had a limited effect (Supplementary

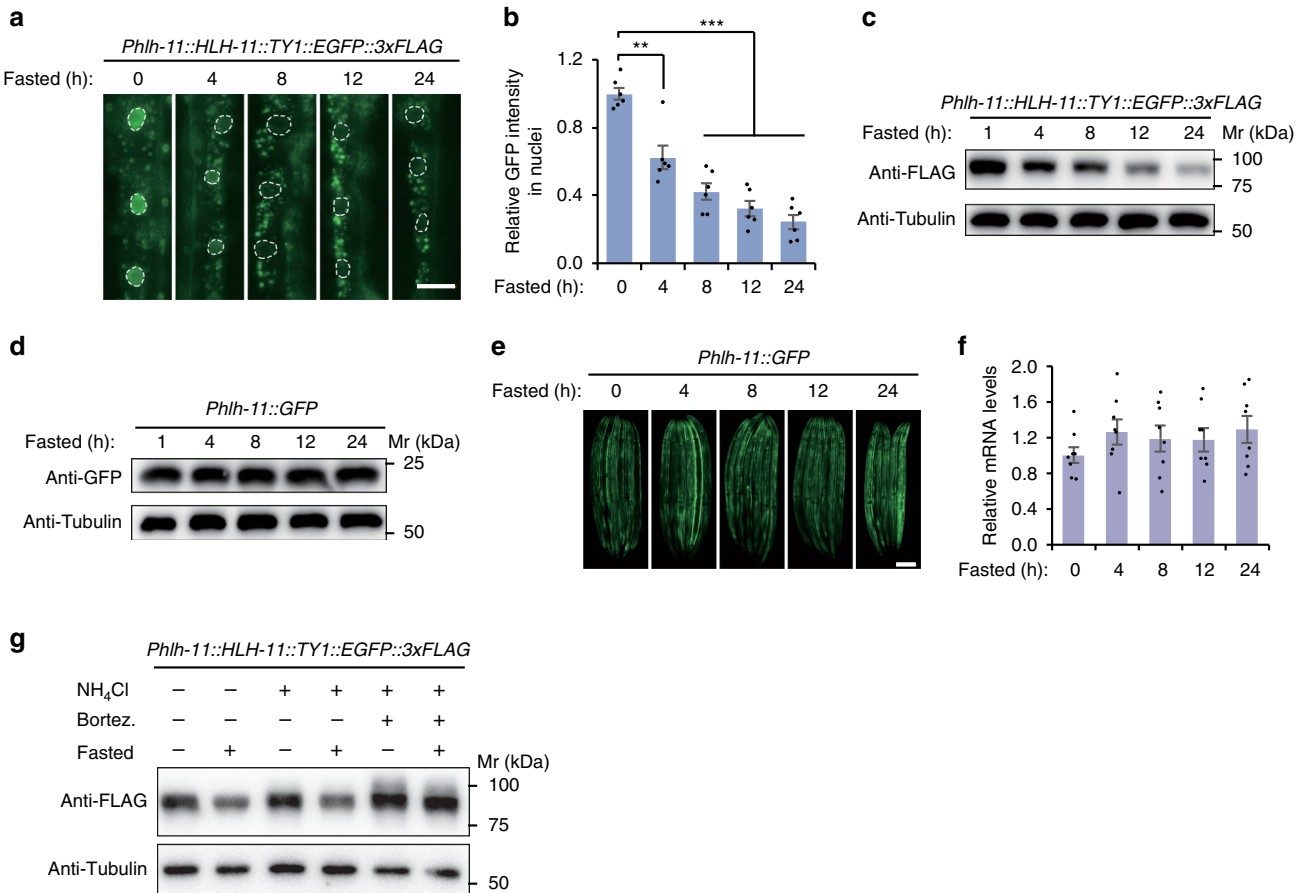

**Fig. 5 HLH-11 protein degradation occurs in response to starvation. a** Representative fluorescence images showing upper intestines of worms expressing *Phlh-11::HLH-11::TY1::EGFP::3xFLAG* fasted for the indicated time period. Three independent experiments were performed with similar results. Scale bar, 20 μm. **b** Quantification of nuclear GFP intensity of worms fasted in **a**. Six nuclei were measured under each condition. **$p = 0.0013$ (4 vs 0 h); ***$p = 5.3E-6$ (8 vs 0 h), ***$p = 4.5E-7$ (12 vs 0 h), ***$p = 1.3E-7$ (24 vs 0 h). **c** Representative immunoblotting images showing the protein levels of HLH-11 and Tubulin (loading control) in *Phlh-11::HLH-11::TY1::EGFP::3xFLAG* worms fasted for the indicated period of time. Three independent experiments were performed with similar results. **d** Representative immunoblotting images showing the protein levels of GFP and Tubulin (loading control) in *Phlh-11::GFP* worms fasted for the indicated period of time. Three independent experiments were performed with similar results. **e** Representative fluorescence images of *Phlh-11::GFP* worms fasted for the indicated time period. $n = 5-6$ worms per condition. Three independent experiments were performed with similar results. Scale bar, 100 μm. **f** qPCR analyses of endogenous *hlh-11* transcript levels in wild-type worms fasted for the indicated period of time. $n = 8$ biologically independent samples. **g** Representative immunoblotting images showing the protein levels of HLH-11 and Tubulin (loading control) in worms treated with or without the proteasome inhibitor Bortezomib (5 μg/ml) or the lysosome inhibitor $NH_4Cl$ (200 mM). Three independent experiments were performed with similar results. Error bars indicate mean ± SEM. Statistical analyses were performed by Student's *t* test (unpaired, two tailed), **$p < 0.01$; ***$p < 0.001$. Source data are provided as a Source data file.

Fig. 5f, g). Altogether, these results suggest that nutrient deprivation activates HLH-11 protein degradation in a manner that may depend on TOR signaling.

**TFAP4 is a mammalian homolog of HLH-11 that regulates lipid metabolism.** Due to the essential function of nutrient sensing and the starvation response in organismal survival, we reasoned that the function and regulation of HLH-11 will be under selective pressure and therefore conserved across species. Indeed, HLH-11 is evolutionarily conserved across phyla (Supplementary Fig. 6). TFAP4, the predicted human homolog of HLH-11, shares 35% overall identity with *C. elegans* HLH-11, which increases to 80% in the HLH domain. To test the function of human TFAP4 in *C. elegans*, we first optimized its coding sequence using *C. elegans* Codon Adaptor[37]. Expression of this optimized TFAP4, driven by the intestine-specific promoter *gly-19*, suppressed the induction of the *Plipl-3::gfp* or *Pacs-2::gfp* reporter in *hlh-11* KO strains (Fig. 6a, b). These results confirm that TFAP4 is the bona fide homolog of HLH-11 in mammals. Furthermore, transcription of a luciferase

reporter driven by the *lipl-3* or *acs-2* promoter was repressed by TFAP4 and HLH-11 in a mammalian system (Fig. 6c), which indicates that TFAP4/HLH-11 possesses the function of a transcriptional repressor.

We next investigated the function of TFAP4 to see whether it plays an evolutionarily conserved role in modulating lipid metabolism in response to nutrient availability in higher eukaryotes. Similar to *C. elegans* HLH-11, the protein level of TFAP4 in mammalian cells also responded to nutrient availability. Starvation of HepG2 cells with Hanks' Balanced Salt Solution (HBSS) revealed a time-dependent reduction of TFAP4 protein (Fig. 6d). Reduction of TFAP4 protein level during fasting also occurs via a post-transcriptional mechanism, because transcript levels of TFAP4 were actually upregulated upon starvation (Fig. 6e). We then treated HepG2 cells with the lysosomal inhibitor Chloroquine (CQ) or the proteasome inhibitor MG132 and tested whether TFAP4 is also regulated by lysosome- or proteasome-mediated protein degradation in response to nutrient availability. Reduction of TFAP4 protein level in starved cells was

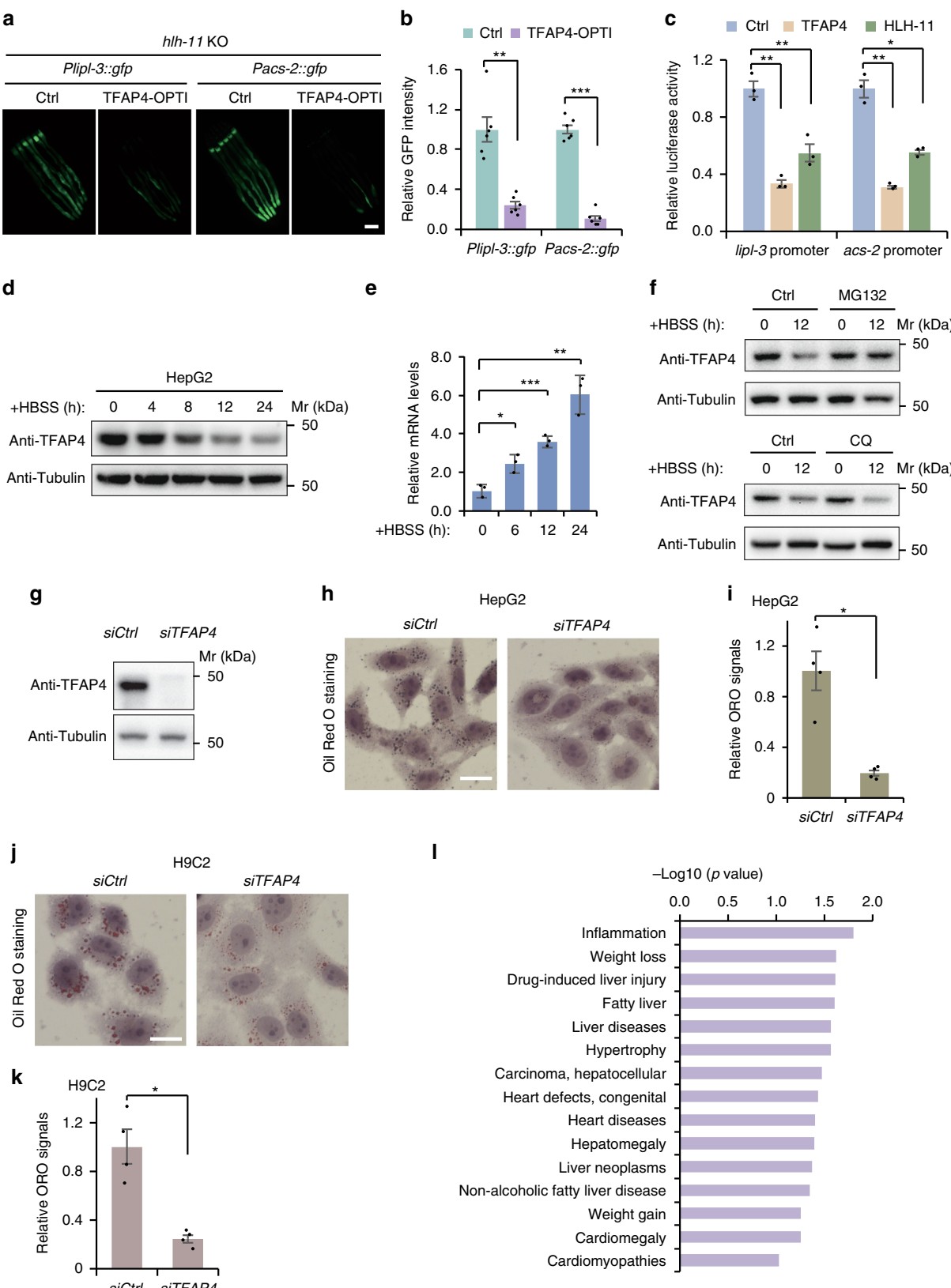

suppressed by MG132 but not by CQ treatment (Fig. 6f), which suggests that TFAP4 is degraded through the proteasomal pathway when nutrients are depleted. We also found that TFAP4 levels also affect fat storage in mammalian cells. Knockdown of

TFAP4 by small interfering RNA (siRNA) significantly reduced the ORO staining intensity in HepG2 liver cells (Fig. 6g–i) or H9C2 cardiac muscle cells (Fig. 6j, k). Lastly, we analyzed the databases[38,39] for diseases associated with TFAP4 expression and

**Fig. 6 TFAP4, the mammalian homolog of HLH-11, plays a conserved role in regulating lipid metabolism. a** Representative fluorescence images of *Plipl-3::gfp; hlh-11* KO or *Pacs-2::gfp; hlh-11* KO worms with exogenous expression of codon-optimized TFAP4 driven by the intestine-specific promoter *gly-19*. Three independent experiments were performed with similar results. Scale bar, 100 μm. **b** Quantification of GFP intensity in **a**. $n = 6$ worms examined per condition. **$p = 0.0013$; ***$p = 1.1E−7$. **c** Transcriptional activity assay of TFAP4 and HLH-11. $n = 3$ biologically independent samples. The exact $p$ values are provided in the Source data. **d** Immunoblotting to detect endogenous protein levels of TFAP4 and Tubulin (loading control) in HepG2 cells starved in HBSS for the indicated period of time. Three independent experiments were performed with similar results. **e** qPCR analyses of TFAP4 transcript levels in HepG2 cells starved in HBSS for the indicated period of time. $n = 3$ biologically independent samples. *$p = 0.017$; ***$p = 0.0006$; **$p = 0.007$.
**f** Immunoblotting to detect endogenous protein levels of TFAP4 and Tubulin (loading control) in HepG2 cells starved in HBSS for the indicated period of time. Cells were pretreated with MG132 (20 μg/ml) or Chloroquine (CQ, 50 μM) for 2 h before starvation. Three independent experiments were performed with similar results. **g** Immunoblotting to detect endogenous protein levels of TFAP4 and Tubulin (loading control) in HepG2 cells transfected with control siRNA or TFAP4 siRNA. Three independent experiments were performed with similar results. **h, j** Representative Oil Red O staining images of HepG2 (**h**) or H9C2 (**j**) cells transfected with control siRNA or TFAP4 siRNA. Scale bar, 20 μm. **i, k** Quantification of Oil Red O staining of HepG2 or H9C2 cells. $n = 4$ biologically independent samples. *$p = 0.0131$ (**i**); *$p = 0.0102$ (**k**). **l** Diseases predicted to be associated with TFAP4 expression. The values of $−\log10(p$ value) were obtained from the Harmonizome database (https://maayanlab.cloud/Harmonizome/gene/TFAP4). Error bars indicate mean ± SEM. Statistical analyses were performed by Student's *t* test (unpaired, two tailed), *$p < 0.05$; **$p < 0.01$; ***$p < 0.001$. Source data are provided as a Source data file.

found that "Inflammation", "Fatty Liver" and "Liver Diseases" are among the top diseases (Fig. 6l). Taken together, these observations indicate that TFAP4, the mammalian homolog of HLH-11, plays a conserved role in regulating lipid metabolism in response to nutrient availability.

## Discussion

A successful strategy for organisms to cope with nutrient fluctuation requires accurate sensing of food availability and rapid adjustment of metabolic programs. In this study, we show that HLH-11 acts as a nutrient-sensitive transcription repressor that promotes fat utilization to meet energy demands under nutrient-deprived conditions. Through mRNA-seq analysis, we noticed that inactivation of HLH-11 mimics a "dietary-restricted state", even during ad libitum feeding, to activate transcription of genes encoding enzymes that catalyze multiple steps of lipid catabolism. These include lysosomal lipases that break down body fats in lipophagy and ACS and ACDH involved in fatty acid β-oxidation. The coordinated transcriptional response of genes that affect multiple steps in lipid catabolism allows precise control of lipid breakdown and reprogramming of the lipid landscape in *C. elegans*. More importantly, we show that TFAP4, the mammalian homolog of HLH-11, plays an evolutionarily conserved role in regulating fat metabolism. Given that overnutrition represents one of the major risk factors for metabolic diseases, including obesity, diabetes, and cardiovascular diseases, approaches to mimicking a "dietary-restricted state" might be important for avoiding excess weight and obesity and could be beneficial to body weight control and healthy aging. Therefore, inactivation of HLH-11, which induces metabolic changes resembling dietary restriction, may have therapeutic applications in the fight against lipid metabolism diseases.

Transcriptional responses represent a major regulatory mechanism for executing metabolic reprogramming to withstand food limitation. In line with this notion, several transcription factors and nuclear hormone receptors have been reported to mediate starvation responses in *C. elegans*[4,15], including both transcription activators and suppressors. A well-received "competition model" has been proposed[15,40]: transcription of lipid catabolism genes is repressed by bHLH-like transcription factors under well-fed conditions, and during starvation, the downregulation of transcriptional repressors allows accessibility of transcriptional activators to the promoter regions of lipid catabolism genes, thereby activating fat utilization. For example, in *C. elegans*, the transcription factor MXL-3 has been reported to repress lipase genes when food is available, and the downregulation of MXL-3 releases the binding site, allowing the

association of HLH-30 with lipase gene promoters to activate their transcription. Interestingly, TFEB, the mammalian homolog of HLH-30, plays a conserved role in mediating lipid metabolism and is regulated through a similar mechanism[3]. However, MAX, the mammalian homolog of MXL-3, is not involved in nutrient sensing and fat utilization, which suggests the existence in higher eukaryotes of other factors that repress lipase gene transcription under ad libitum feeding conditions[15]. TFAP4, identified in our study, may represent such a factor. Interestingly, TFAP4 and TFEB both associate with the E-box motif (CANNTG)[41–43], which points toward the possibility that these two transcription factors may constitute an antagonistic pair to compete for the binding sites in promoters of lipid catabolism genes.

Our results unravel a transcriptional response that is able to sense nutrient fluctuation. However, it remains to be determined how nutrient signals are coupled to the transcription factor HLH-11. We find that, in response to food deprivation, *C. elegans* HLH-11 protein is degraded through both lysosomal and proteasomal pathways. The nutrient sensor mTOR, but not AMPK, seems to regulate HLH-11 protein levels. Interestingly, mTOR has been shown to modulate the induction of *C. elegans* HLH-30 and the induction and subcellular localization of its mammalian homolog TFEB during nutrient starvation[3,15]. This suggests that a transcription repressor and activator are coordinately regulated through the same nutrient-sensing pathway. It should also be noted that MXL-3, another transcription repressor of lipase genes reported in *C. elegans*, activates an acute response to nutrient deprivation[15]. *mxl-3* transcript levels were greatly reduced after 5 h of fasting but became elevated 12 h after food withdrawal. Regulation of *mxl-3* does not depend on *Ce*.TOR. By contrast, we find that HLH-11 may mediate a prolonged response to nutrient deficiency. Degradation of HLH-11 protein is observed even after 24 h of fasting. Altogether, these observations point toward complex regulation of the transcriptional response when nutrient levels fluctuate.

## Methods

**Worm strains and maintenance**. *C. elegans* strains were grown on standard nematode growth medium (NGM) plates with *Escherichia coli* OP50 at 20 °C. N2, *unc-119(ed3); wwEx38[Phlh-11::GFP; unc-119(+)]*, *unc-119(tm4063); wgIs396 [Phlh-11::hlh-11::TY1::EGFP::3xFLA+unc-119(+)]* and *hlh-11(ok2944)* were obtained from the *Caenorhabditis* Genetics Center (CGC). The following strains were generated in our lab. YSL11(*liuIs3[Plipl-3::gfp; Podr-1::dsRed]*), YSL12(*liuIs4 [Pacs-2::gfp; Podr-1::dsRed]*), YSL13(*liuIs5[Phlh-11::hlh-11::gfp; Podr-1::dsRed]*), YSL14 *hlh-11(ko1)*, YSL15(*liuIs3[Plipl-3::gfp; Podr-1::dsRed]; hlh-11(ko1)*), YSL16 (*liuIs4[Pacs-2::gfp; Podr-1::dsRed]; hlh-11(ko1)*), YSL17(*liuEx2[Phlh-11::hlh-11:: mCherry]; liuIs11[Plipl-3::gfp; Podr-1::dsRed]; hlh-11(ko1)*), YSL18(*liuEx2[Phlh-11:: hlh-11::mCherry]; liuIs12[Pacs-2::gfp; Podr-1::dsRed]; hlh-11(ko1)*), YSL19(*liuEx3 [Phlh-11::hlh-11(ΔHLH)::mCherry]; liuIs11[Plipl-3::gfp; Podr-1::dsRed]; hlh-11 (ko1)*), YSL20(*liuEx3[Phlh-11::hlh-11(ΔHLH)::mCherry]; liuIs12[Pacs-2::gfp; Podr-*

1::dsRed]; hlh-11(ko1)), YSL21(liuEx4[Pgly-19::hlh-11::mCherry]; liuIs12[Pacs-2:: gfp; Podr-1::dsRed]; hlh-11(ko1)), YSL22(liuEx5[Prab-3::hlh-11::mCherry; liuIs12 [Pacs-2::gfp; Podr-1::dsRed]; hlh-11(ko1)), YSL23(liuEx6[Ppgp-12::hlh-11::mCherry]; liuIs12[Pacs-2::gfp; Podr-1::dsRed]; hlh-11(ko1)).

**Cell culture**. HepG2 cells were cultured in Eagle's Minimum Essential Medium (ATCC, #30-2003) supplemented with 10% fetal bovine serum (FBS; Gibco # 10099-141) and penicillin and streptomycin (Hyclone, #SV30010) at 37 °C. HEK293T and H9C2 cells were cultured in Dulbecco's Modified Eagle Medium (Hyclone, #SH30243.01) supplemented with 10% FBS (Biowest, #S181B-500) and penicillin and streptomycin (Hyclone, #SV30010) at 37 °C. HBSS starvation was performed by changing culture medium to HBSS (Gibco, #14025-092), and cells were incubated in HBSS for the indicated period of time before sample preparation.

**Fluorescence microscopy**. Worms were immobilized in M9 with 50 mM sodium azide, mounted on 2% agarose pads, and imaged using a Zeiss Axio Imager M2 microscope. Images were acquired using the Software Zen 1.1.2.0 (Zeiss). Comparable images were captured with the same exposure time and magnification. Quantifications of GFP in reporter worms were performed using the ImageJ software to calculate integrate or mean intensity.

**Reverse transcription–qPCR**. Synchronized young adult worms were washed off plates with M9 and resuspended in TRIzon Reagent (CWBIO, #CW0580). Total RNA was isolated by chloroform extraction, followed by isopropanol precipitation. cDNA was prepared using One-step gDNA Removal and cDNA Synthesis SuperMix (Transgen Biotech, #AT311). RT-qPCR was performed using SYBR Green PCR Master Mix (Bio-Rad, #1725121) to measure the expression levels of target genes. For quantification, mRNA levels were normalized to ama-1.

Primer sequences for RT-qPCR are provided in Supplementary Table 1.

**RNAi screening**. RNAi bacteria were cultured overnight in LB medium with 50 μg/ml Carbenicillin at 37 °C and then seeded onto RNAi agar plates containing 1 mM isopropyl-β-D-thiogalactoside. The plates were left to dry in a laminar flow hood and incubated at room temperature overnight to induce double-stranded RNA expression. Synchronized Plipl-3::gfp and Pacs-2::gfp hatchlings were seeded onto the plates and cultured at 20 °C for 48 h before examination for GFP signal.

**ORO staining**. Synchronized worms were collected, washed 3 times with phosphate-buffered saline (PBS), and fixed for 1 h in MRWB buffer (80 mM KCl, 20 mM NaCl, 7 mM Na$_2$EGTA, 0.5 mM spermidine-HCl, 0.2 mM spermine, 15 mM Na-PIPES pH 7.4, 0.1% β-mercaptoethanol) containing 2% paraformaldehyde (PFA). Worms were then washed with PBS to remove PFA, resuspended in 60% isopropanol, and incubated at room temperature for 15 min to dehydrate them. A 60% ORO solution was freshly prepared by diluting ORO Solution (Sigma, #O1391) with water, rocking overnight and filtering with a 0.22-μm filter. After removal of isopropanol, worms were incubated with rocking in 60% ORO solution overnight at room temperature. Stained worms were mounted onto 2% agarose pads, and imaged using a Zeiss Axio Imager M2 microscope. Images were acquired using the Software Zen 1.1.2.0 (Zeiss). To quantify ORO signals, images were first converted to RGB color. The mean intensity in the green channel on the upper intestine below the pharynx of each animal was calculated using the Fiji software. For mammalian cells, cell culture medium was removed and cells were washed with PBS once, followed by fixation with 4% PFA for 30 min. Cells were washed once again with PBS and incubated in 60% ORO working solution for 2 h. After removal of ORO solution, cells were then washed twice with PBS. Nuclei was stained with hematoxylin staining solution (BBI Life Sciences, #E607317) before images were taken. Images were acquired using the Software Zen 1.1.2.0 (Zeiss). For quantification, cells were first washed 5 times with 60% isopropanol. To extract ORO, 100% isopropanol was added, then 80% of the extraction volume was removed, and the ORO content was assessed by measuring the absorbance at 510 nm.

**Nile Red staining**. Nile Red (Molecular Probes, #LSN1142) was dissolved in acetone and 1 mg/ml stock solution was kept at 4 °C. Stock solution was freshly diluted in PBS to 1 μg/ml and 0.5 ml of the diluted solution was added to the surface of NGM plates (~11 ml agar) seeded with OP50. Synchronized L1 worms were allowed to develop on the plates until the adult stage. Day 1 worms were mounted onto 2% agarose pads and imaged using a Zeiss Axio Imager M2 microscope. Images were acquired using the Software Zen 1.1.2.0 (Zeiss). Quantifications of Nile Red signals were performed using the ImageJ software to calculate integrated intensity.

**mRNA-sequencing**. For each sample, approximately 2000 synchronized young adult worms were collected and resuspended in TRIzon Reagent (CWBIO, #CW0580). Total RNA was isolated by chloroform extraction, followed by isopropanol precipitation. Sequencing was performed using a HiSeq2500 sequencing system (Illumina). The clean reads were aligned to the WBcel235 genome assembly with TopHat2. Differentially expressed genes were then identified using DEseq2 (an R package for differential expression analysis)[44]. p Values were calculated by two-sided Wald test and corrected for multiple testing using the Benjamini–Hochberg method. Genes with an adjusted p value <0.05 were selected as differentially expressed genes. Induced (KO_up or OE_up) or suppressed (KO_down or OE_down) genes were obtained by comparing transcript levels in hlh-11 KO or OE worms to wild-type animals. For the functional classifications, p values were calculated by one-sided Fisher's exact test modified by the DAVID (Database for Annotation, Visualization, and Integrated Discovery) tool[45] and corrected for multiple testing using the Bonferroni method. Fold changes of transcript levels of genes involved in lipid metabolism, glycolysis, or the TCA cycle were calculated by dividing the expression level of each gene in KO or OE worms by that of the wild-type worms. For each condition, two replicates were sequenced and analyzed. The data sets were deposited in Gene Expression Omnibus with the accession number GSE140866.

**Lipid profiling**. For each sample, approximately 10,000 day 1 adult worms were collected and washed three times with M9 in a 15-ml tube and rotated for 30 min at room temperature to get rid of gut bacteria. After gut clearance, worms were washed twice with M9 and centrifuged. Pellets were frozen in liquid nitrogen, then kept at −80 °C until processed. Lipid extraction and liquid chromatography/mass spectrometry (LC/MS) analysis were performed by LipidALL Technologies Co., Ltd. (www.lipidall.com). Lipids were extracted twice using chloroform:methanol (1:2), with an internal standard added. LC/MS analysis was carried out by UPLC-QTRAP 6500 PLUS (Sciex). Four replicates from each condition were used for analyses. Statistical analysis was performed using one-sided Tukey's honestly significant difference test.

**Starvation assays**. L4 animals were washed off plates and rinsed three times with M9. Half of the worms were then seeded back onto plates with food, and the other half were seeded onto plates without food. Worms were then cultured for each indicated period of time.

**Bortezomib and NH$_4$Cl treatment**. L4 worms were washed off plates with M9 and collected into tubes containing M9 and OP50. Bortezomib (5 μg/ml) or NH$_4$Cl (200 mM) was added into the M9. The tubes were rotated at 20 °C for 6 h (Bortezomib) or 3 h (NH$_4$Cl). Worms were then equally divided into two tubes, one with food and the other one without food, and rotated for 12–16 h before they were collected and prepared for immunoblotting.

**Immunoblotting**. For C. elegans samples, worms were collected and rinsed 2–3 times with M9. After the final rinse, a certain volume of M9 was left and ¼ volume of 5× SDS sample buffer (312.5 mM Tris-HCl pH 6.8, 12.5% sodium dodecyl sulfate (SDS), 0.01% Bromophenol Blue, 25% β-mercaptoethanol, 50% glycerol) was added. The samples were then heated at 95 °C for 10 min before loading onto SDS-polyacrylamide gel electrophoresis (PAGE) gels. For cell samples, after washing with PBS, RIPA buffer (20 mM Tris-HCl, 150 mM NaCl, 1 mM Na$_2$EDTA, 1 mM EGTA, 1% NP-40, 1% sodium deoxycholate, 2.5 mM sodium pyrophosphate, 1 mM β-glycerophosphate) with EDTA-free Proteinase Inhibitor Cocktail (Roche) and 1 mM dithiothreitol (DTT) was added into the cell culture plates. Plates were incubated on ice for 10 min. Cell lysates were then collected into tubes and ¼ volume of 5× SDS sample buffer was added. The samples were heated at 95 °C for 10 min before loading onto SDS-PAGE gels. Antibodies used were anti-GFP antibody (diluted 1:1000, Sungene Biotech, #KM8009), anti-FLAG antibody (diluted 1:1000, Sigma, #F1804), anti-TFAP4 antibody (diluted 1:1000, Santa Cruz, #377042), and anti-Tubulin antibody (diluted 1:1000, Abcam, #6161).

**Chromatin immunoprecipitation–qPCR**. Synchronized worms were raised until the young adult stage, then collected in a 15-ml tube and washed 3–5 times with PBS. Worms were rotated in PBS for 30 min at room temperature to get rid of gut bacteria, then washed 3 times. After that, worms were rotated in PBS containing 2% formaldehyde for 40 min and washed 3 times with PBS again. To enrich HLH-11–DNA complexes, the nuclei were separated and lysed in ChIP lysis buffer (50 mM HEPES-KOH pH 7.5, 150 mM NaCl, 1 mM EDTA pH 8.0, 0.1% sodium deoxycholate, 1% Triton X-100, 0.1% SDS) containing 1 mM PMSF, 1 mM DTT, and EDTA-free Proteinase Inhibitor Cocktail (Roche). After sonication and centrifugation, the supernatant was incubated with GFP antibody (5 μg/ChIP reaction, Abcam, #ab290) at 4 °C overnight and collected with Dynabeads Protein G (Invitrogen 10004D). After extensive washes, immunocomplexes were treated with proteinase K (ZYMO, #D3001-2-B). Bound DNA in the precipitates, as well as input DNA, were extracted, purified, and subjected to qPCR to detect the enrichment of targets.

**Transcriptional activity assay**. The promoter of lipl-3 or acs-2 was cloned into pGL4.17 vector. pGL4.17 plasmids carrying the lipl-3 or acs-2 promoter were co-transfected with pRL-CMV (internal control) into HEK293T cells, together with plasmids expressing TFAP4 or HLH-11. Thirty-six hours after transfection, transcriptional activities were measured with the dual luciferase reporter assay system (Promega, #E1910).

**Statistics and reproducibility**. Statistical analyses were performed by Student's *t* test (unpaired, two tailed) with at least three replicates, unless otherwise indicated. Statistical analyses were performed in Excel (Microsoft) or GraphPad Prism (GraphPad Software Inc.). Error bars indicate mean ± SEM unless otherwise indicated. Experiments yielding quantitative data for statistical analyses were performed independently at least three times with similar results. Micrographs and immunoblotting images are representative images of at least two independent experiments, with similar results.

**Reporting summary**. Further information on research design is available in the Nature Research Reporting Summary linked to this article.

## Data availability

mRNA-seq data reported in this paper have been deposited in the NCBI Gene Expression Omnibus (GEO) database under accession code GSE140866. −Log10(*p* value) of TFAP4's association with diseases were obtained from Harmonizome (https://maayanlab.cloud/Harmonizome/gene/TFAP4). Other data supporting the findings of this study are available from the authors upon reasonable request. Source data are provided with this paper.

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

## Acknowledgements

We thank the *Caenorhabditis* Genetics Center (CGC) for providing strains. We are grateful to Dr. Isabel Hanson for reading and editing the manuscript. Y. Liu was supported by grants from the National Natural Science Foundation of China (grant nos. 91854205 and 31925012), the Ministry of Science and Technology of China (National Key Research and Development Program of China grant no. 2017YFA0504000973), and an HHMI International Research Scholar Program (grant no. 55008739). This work was also supported by Beijing Advanced Innovation Center for Genomics and Peking-Tsinghua Center for Life Sciences. C.-Y.L. was supported by grants from the Ministry of Science and Technology of China (National Key Research and Development Program of China grant no. 2019YFA0801801 and no. 2018YFA0801405) and the National Natural Science Foundation of China (grant no. 31871272). Y. Liu acknowledges support from the Tencent Foundation through the XPLORER PRIZE.

## Author contributions

Y. Li and Y. Liu conceived the study. Y. Li carried out all the experiments. W.D. and C.-Y.L. performed RNA-seq analyses. Y. Li and Y. Liu wrote the manuscript.

## Competing interests

The authors declare no competing interests.
