## [Peer Review File · Nature Communications]

Reviewers' comments:

Reviewer #1 (Remarks to the Author):

In this study, the authors found that the transcription factor HLH-11 acts to negatively regulate lipid catabolism genes during periods of food availability in *C. elegans*. They made this discovery by screening using reporter strains of two lipid catabolism genes that are up-regulated during fasting/starvation. The authors showed that in the *hlh-11* KO worms under well-fed conditions, there is less fat accumulation, and lower levels of various TAG and DAG lipid species. Interestingly, the authors showed that after food deprivation, the HLH-11 protein is degraded via proteasomes and lysosomes, thus releasing the inhibition of the lipid catabolism gene transcription. Importantly, the findings appear to be relevant to the human homolog TFAP4. The findings are novel, and this study will be of significant interest and importance to the field. Modulation of this type of transcription factor could be a potential way to treat obesity.

Overall, I found the paper to be well written and the experiments carried out carefully. I just have a couple of concerns. No page numbers make this a bit difficult, page numbers should be included in future submissions.

1. In the first paragraph of the results, the statement, "...acs-2, encoding an acyl- coA synthetase that catalyzes fatty acid beta-oxidation..." is not precise. The ACS proteins catalyze the activation of a fatty acid to a fatty acyl-CoA. Please change the sentence to make it more accurate and precise.

2. In Figure 1, the photo shown in Fig 1D indicate no visible GFP in the *acs-2::GFP* worms raised with adequate food. Contrastingly, the photo shown in Fig 1F, bottom left corner, shows green worms, which are also *acs-2::GFP* raised with adequate food. Perhaps the photo in Fig 1D could be changed to indicate some GFP under well fed conditions. I realize this is all relative, and that careful quantification was done, but visually the way it is presented here seems to indicate variability in the experiment that wasn't addressed in the text.

3. In the Results text, in the paragraph describing Figure 2, "...nuclei of intestinal cells, neurons and excretory cells (wormbase, Fig. 2a). These cells are located in major endocrine tissues that coordinate *C. elegans* metabolism..."

I'm not sure if it is correct to refer to those as "endocrine tissue". Obviously those cells carry out many other functions other than endocrine. In *C. elegans* the major tissues are skin, intestine, muscle, neurons, germline... No specific endocrine tissues. Maybe the wording could be something like, "cells performing endocrine functions..."

4. For the tissue-specific rescue experiments, I understand that the three promoters chosen were based on the expression pattern exhibited by *hlh-11*. However, recently the skin has been shown to be very important in lipid metabolism. I think the claim, "HLH-11 may act cell-autonomously to regulate fat metabolism.." is speculative, given that it wasn't tested whether HLH-11 expression in the skin could affect lipid catabolism.

5. In the first section of the Results, the authors stated that “..the cis-regulatory motif of HLH- 11 has been found in the promoter of atgl-1, a triglyceride lipase..” This makes me wonder if this motif is present on the promoter of acs-2, lipl-3, or other lipid catabolism genes that were examined in this study.

Reviewer #2 (Remarks to the Author):

Review of

HLH-11 modulates lipid metabolism in response to nutrient

Availability

NComm

This manuscript focuses on HLH -11 and its role in lipid the towel is it a lipid metabolism in the face of starvation. Overall the experiment presented are adequate but does ;ittle to advance the field-the manuscript reads more as a description of the hlh-11 mutant and overexpression phenotypes. Therefore, this manuscript is more suitable for another journal. The following are some pertinent questions that should be addressed:

1-why pick lipl-3 and acs-2? What is there connection? the manuscript is motivated by these genes and their functions. Why use on or both for validation?

2- from the RNAi screen, the authors go 57 hits then focus on hlh-11. Why? Did the homology have anything to do with it?

3- It is possible that this is all due to tax-6- please discuss.

Lee et al.

Mol Cells. 2009 Nov 30;28(5):455-61. Identification and characterization of a putative basic helix-loop-helix (bHLH) transcription factor interacting with calcineurin in *C. elegans*.

Wang et al. (2017)

Molecules. 2017 Jun 26;22(7)

Calcineurin Antagonizes AMPK to Regulate Lipolysis in *Caenorhabditis elegans*.

Reviewer #3 (Remarks to the Author):

The key message of the manuscript by Yi Li and colleagues is that starvation results in degradation of HLH-11, a transcription factor. This then brings about a transcriptional change that ultimately promotes fat utilization through enhanced beta oxidation. Overall, the paper is well written and the data are presented in a logical order. There are a number of attractive features to the manuscript and many of the experiments are well performed. However, the conclusions drawn by the author fail to take into consideration a fundamental aspect of lipid metabolism in hermaphroditic *C. elegans* (see below for detail). It is worth noting that lack of consideration of this fundamental issue is applicable to the vast majority of papers pertaining to lipid analysis in *C. elegans*.

Major comments:

1) While it is certainly true that *C. elegans* initiate fat breakdown for energetic purposes upon starvation, it is equally true that upon starvation gravid *C. elegans* hermaphrodites massively transfer lipids to their developing progeny in the form of yolk. Yolk is made in the intestine, the very same tissue that is the focus of the investigation in this manuscript. None of the fat measurement techniques used in this manuscript specifically discriminate among lipids that are in storage compartments from those that are in yolk. Since yolk formation requires building of lipoprotein particles, it would be expected that a process such as starvation that promotes build-up and transfer of yolk into progeny would be accompanied by major transcriptional changes in lipid metabolic genes. While burning of lipids to generate energy and remodeling of lipids to make and transfer yolk both involve lipid metabolism genes, they have completely different physiological meanings and cannot be lumped together. The authors should, therefore, provide evidence to distinguish whether the effect seen upon *hlh-11* inactivation pertain to lipid utilization for energetic demands of the mother (as they suggest) or to transfer of lipids from the mother to embryos which then exit the mother as eggs laid. For example, what happens to lipid content in males or in larval animals deficient in *hlh-11*?

2) Figure 5 (effects of starvation on stability of the HLH-11 protein): The data, as shown, are convincing. The issue is that the experiments in Figure 5 utilize L4 (e.g. non-gravid) animals while much of the rest of the data (RNAseq, lipid analysis) are conducted on gravid adults. This may seem like a small thing but once again given the enormous effects of the onset of reproduction on lipid metabolism in adult hermaphrodites, these developmental states cannot be considered as equivalent. As suggested above, at least the fat measurements should be done in L4 (ensuring that the animals have not yet committed to generation of embryos).

Other comments:

1) The authors state: "During lipid catabolism, after breakdown of body fats by lysosomal lipases to generates free fatty acids (FFAs), the carbon-carbon bonds of FFAs are further broken down through beta oxidation in mitochondria". In mammals, fatty acids liberated from lipid droplets are directly fed into peroxisomal and mitochondrial oxidation pathways. Lysosome play a role in fats that are imported into cells through lipoproteins. If the intestine is the major site of fat storage, then why are the lipids going through lysosomes prior to being sent to beta oxidations pathways? Can you please clarify?

2) Back to the yolk issue: of course, *acs* and *acdh* genes can play roles in fat oxidation, but they can also play roles in virtually every other aspect of lipid metabolism, for example, yolk formation. Thus, in the absence of direct evidence, the speculations regarding the roles of these genes should be toned down and the appropriate caveats explicitly stated.

3) While it is perfectly reasonable to bring up the potential interplay of lipid metabolic and innate immunity genes, without directly testing the effects of *hlh-11* on innate immunity, the interpretations should be presented with the appropriate caveats.

4) *Glo-4* also affects gut granules, which are lysosome related organelles. Thus, it cannot be concluded that the effects seen in *glo-4* mutants are necessarily due to lysosomes.

Reviewers' comments:

Reviewer #1 (Remarks to the Author):

In this study, the authors found that the transcription factor HLH-11 acts to negatively regulate lipid catabolism genes during periods of food availability in *C. elegans*. They made this discovery by screening using reporter strains of two lipid catabolism genes that are up-regulated during fasting/starvation. The authors showed that in the *hlh-11* KO worms under well-fed conditions, there is less fat accumulation, and lower levels of various TAG and DAG lipid species. Interestingly, the authors showed that after food deprivation, the HLH-11 protein is degraded via proteasomes and lysosomes, thus releasing the inhibition of the lipid catabolism gene transcription. Importantly, the findings appear to be relevant to the human homolog TFAP4. The findings are novel, and this study will be of significant interest and importance to the field. Modulation of this type of transcription factor could be a potential way to treat obesity. Overall, I found the paper to be well written and the experiments carried out carefully. I just have a couple of concerns. No page numbers make this a bit difficult, page numbers should be included in future submissions.

We appreciate the reviewer's interest in our work and his/her thoughtful consideration of our manuscript. Page numbers are now included in the revised submission.

1. In the first paragraph of the results, the statement, "...acs-2, encoding an acyl- coA synthetase that catalyzes fatty acid beta-oxidation..." is not precise. The ACS proteins catalyze the activation of a fatty acid to a fatty acyl-CoA. Please change the sentence to make it more accurate and precise.

We thank the reviewer for this suggestion. We have changed the sentence to "acs-2, encoding an acyl-coA synthetase that catalyzes the activation of fatty acids to fatty acyl-coAs for beta-oxidation".

2. In Figure 1, the photo shown in Fig 1D indicate no visible GFP in the *acs-2::GFP* worms raised with adequate food. Contrastingly, the photo shown in Fig 1F, bottom left corner, shows green worms, which are also *acs-2::GFP* raised with adequate food. Perhaps the photo in Fig 1D could be changed to indicate some GFP under well fed conditions. I realize this is all relative, and that careful quantification was done, but visually the way it is presented here seems to indicate variability in the experiment that wasn't addressed in the text.

We appreciate the point raised by the reviewer. Yes, Fig. 1d and 1f were taken under different exposure times. To avoid any confusion, we provided an updated version of this figure, showing relatively similar GFP levels in the *acs-2::GFP* reporter strains in Fig. 1d and 1f. In addition, we used qPCR to measure endogenous transcript levels of *lipl-3* and *acs-2* in *hlh-11* knockout and overexpression animals (Fig. 1h), and we also observed the same phenotype.

3. In the Results text, in the paragraph describing Figure 2, "...nuclei of intestinal cells, neurons and excretory cells (wormbase, Fig. 2a). These cells are located in major endocrine tissues that coordinate *C. elegans* metabolism..." I'm not sure if it is correct to refer to those as "endocrine tissue". Obviously those cells carry out many other functions other than endocrine. In *C. elegans* the major tissues are skin, intestine, muscle, neurons, germline... No specific endocrine tissues. Maybe the wording could be something like, "cells performing endocrine functions..."

We have followed the reviewer's suggestion to revise this sentence to "These cells can perform endocrine functions to coordinate *C. elegans* metabolism in response to nutrient levels."

4. For the tissue-specific rescue experiments, I understand that the three promoters chosen were based on the expression pattern exhibited by *hlh-11*. However, recently the skin has been shown to be very important in lipid metabolism. I think the claim, "HLH-11 may act cell-autonomously to regulate fat metabolism.." is speculative, given that it wasn't tested whether HLH-11 expression in the skin could affect lipid catabolism.

We appreciate the point raised by the reviewer. Yes, we tested these three promoters because we showed in Fig. 2a that expression of HLH-11 was only observed in the nuclei of intestinal cells, neurons and excretory cells. We agree with the reviewer that skin also plays an important role in lipid metabolism. When we revised the manuscript, we tried very hard to express HLH-11 under the control of a skin-specific promoter (*dpy-7*). However, we could not obtain viable worms after microinjection. The F1 transgenic worms died during the embryonic or early larval stage. Therefore, we deleted the sentence claiming that "HLH-11 may act cell-autonomously to regulate fat metabolism". This does not affect the major conclusions of the manuscript.

5. In the first section of the Results, the authors stated that "...the cis-regulatory motif of HLH-11 has been found in the promoter of *atgl-1*, a triglyceride lipase.." This makes me wonder if this motif is present on the promoter of *acs-2*, *lipl-3*, or other lipid catabolism genes that were examined in this study.

We thank the reviewer for raising this point. Yes, the promoters of *acs-2*, *lipl-3*, and several other lipid catabolism genes such as *lipl-1*, *lipl-4*, *acdh-6* (acyl-CoA dehydrogenase) and *F54C8.1* (hydroxyacyl-CoA dehydrogenase) also contain the cis-regulatory motif of HLH-11. In addition, we provided new results in Supplementary Fig. 1f to show that HLH-11 directly binds to the promoters of *acs-2* and *lipl-3*.

Reviewer #2 (Remarks to the Author):

This manuscript focuses on HLH-11 and its role in lipid metabolism in the face of starvation. Overall the experiment presented are adequate but does little to advance the field-the manuscript reads more as a description of the *hlh-11* mutant and overexpression phenotypes. Therefore, this manuscript is more suitable for another journal. The following are some pertinent questions that should be addressed:

We thank the reviewer for the encouraging remarks on our work.

In this study, we reported several principal findings:

- 1) We carried out a RNAi screen to identify that *hlh-11* acts as a nutrient-sensitive transcription repressor, which promotes fat utilization under nutrient deprivation.
- 2) We performed both transcriptome and lipidomic analysis to characterize the genes and lipids regulated by HLH-11. We found that inactivation of HLH-11 mimics a "dietary-restricted state" to activate transcription of genes involved in multiple steps of lipid catabolism, thus reprogramming the lipid landscape in *C. elegans*.
- 3) We further showed that lysosome- and proteasome-mediated degradation of HLH-11 is an essential mechanism for HLH-11 regulation in response to fasting.
- 4) We showed that the mammalian homolog of HLH-11 plays an evolutionarily conserved role in regulating fat metabolism.

In addition, in the revised manuscript, we have performed ChIP-qPCR experiments to show that HLH-11 directly binds to the promoter of *lipl-3* or *acs-2* (Supplementary Fig. 1g). Moreover, we used luciferase reporter assays to show that HLH-11 and its mammalian homolog TFAP4 repress transcription driven by the *lipl-3* or *acs-2* promoter (Fig. 6c). These new results provide mechanistic insights into how HLH-11 acts to regulate the starvation response.

Nature Communications has published many papers describing the “mutant and overexpression phenotype” of one particular gene in *C. elegans*. Here are examples of a few studies published during the last two years:

PMID: 31704915 <https://www.nature.com/articles/s41467-019-13062-z>

(This study reports the *fln-2* mutation that affects lethal pathology and lifespan in *C. elegans*)

PMID: 31316054 <https://www.nature.com/articles/s41467-019-10759-z>

(This study reports the mutant and overexpression phenotype of the *tcer-1* gene in *C. elegans* innate immunity and lifespan regulation)

PMID: 31827090 <https://www.nature.com/articles/s41467-019-13540-4>

(This study reports the *sqst-1* mutation that affects autophagy and lifespan in *C. elegans*)

PMID: 31346165 <https://www.nature.com/articles/s41467-019-11275-w>

(This study reports the *kif-1* RNAi phenotype in xenobiotic response and lifespan in *C. elegans*)

PMID: <https://www.nature.com/articles/s41467-018-06051-1>

(This study reports the *fib-1* RNAi phenotype in pathogen response in *C. elegans*)

Our study not only carried out comprehensive transcriptome and lipidomic analysis, but also revealed the conserved role of the homologous gene in regulating mammalian lipid metabolism. Therefore, we hope that the reviewer finds that the revised manuscript fits the level of a Nature Communications publication.

1-why pick *lipl-3* and *acs-2*? What is there connection? the manuscript is motivated by these genes and their functions. Why use one or both for validation?

We thank the reviewer for raising this question. In the original manuscript, we stated that “fasted animals had significantly increased transcript levels of *lipl-3*, encoding a lysosomal lipase that breaks down lipid-droplet fats to fatty acids, and *acs-2*, encoding an acyl-coA synthetase that catalyzes fatty acid beta-oxidation. Therefore, to identify novel regulators and understand the molecular mechanisms that link nutrient availability to body fat regulation, we generated two starvation-responsive reporter strains in *C. elegans*, *Plip1-3::gfp* and *Pacs-2::gfp*”.

Motivated by the reviewer’s question, we added more sentences in page 4 to better explain our rationale. We have now revised the text to “Consistent with the decreased body fat stores under starved conditions, fasted animals had significantly increased transcript levels of *lipl-3*, encoding a lysosomal lipase that breaks down lipid-droplet fats to fatty acids, and *acs-2*, encoding an acyl-coA synthetase that catalyzes the activation of fatty acids to fatty acyl-CoAs for beta-oxidation. Because the degradation of lipids, in response to fasting, consists of two steps, lipophagy/lipolysis and beta-oxidation, we generated two starvation-responsive reporter strains in *C. elegans*, *Plip1-3::gfp* and *Pacs-2::gfp*. We reasoned that by using both of the reporters, our screen will uncover novel regulators that function in upstream signaling to modulate starvation-induced lipid catabolism.”

We hope the revised description can better explain the reason for picking these two reporters.

2- from the RNAi screen, the authors go 57 hits then focus on *hlh-11*. Why? Did the homology have anything to do with it?

We have provided the reason for working on *hlh-11* in the original manuscript (page 5). “First, RNAi knockdown of *hlh-11* caused relatively high induction of the *lipl-3* and *acs-2* reporters. *hlh-11* RNAi efficiency was verified via both imaging and immunoblotting. Second, *hlh-11* encodes a bHLH transcription factor, which is likely to modulate a large set of downstream genes and play an important role in reprogramming lipid metabolism in response to nutrient deficiency. Third, the cis-regulatory motif of HLH-11 has been found in the promoter of *atgl-1*, a triglyceride lipase that catalyzes lipolysis in response to fasting.”

3- It is possible that this is all due to *tax-6*- please discuss.

Lee et al.

Mol Cells. 2009 Nov 30;28(5):455-61. Identification and characterization of a putative basic helix-loop-helix (bHLH) transcription factor interacting with calcineurin in *C. elegans*.

Wang et al. (2017)

Molecules. 2017 Jun 26;22(7)

Calcineurin Antagonizes AMPK to Regulate Lipolysis in *Caenorhabditis elegans*.

We appreciate the reviewer’s insightful suggestion. It has been reported that HLH-11 associates with TAX-6 and may function downstream of *tax-6* (Lee et al., Molecules and Cells, 2009). When we initially worked on this project, we tested the possibility that HLH-11 and TAX-6 may function in concert to regulate lipid metabolism in response to starvation. However, neither a *tax-6* loss-of-function allele (*ok2065*) nor a *tax-6* gain-of-function allele (*jh107*) affected the nuclear localization of HLH-11. In addition, Wang et al. (2017) showed that the *tax-6* loss-of-function allele (*ok2065*) results in a reduction in body fat level. If HLH-11 functions to regulate *tax-6* transcription, we would expect to see lower *tax-6* expression in *hlh-11* KO worms, which have higher levels of body fat. In contrast, based on our RNA-seq analysis, *hlh-11* KO promoted the transcription of *tax-6*. Therefore, we think the effect of HLH-11 in regulating lipid metabolism may not simply be due to *tax-6*.

In addition, in the revised manuscript, we have performed ChIP-qPCR experiments (Supplementary Fig. 1g) to show that HLH-11 directly binds to the promoter of *lipl-3* or *acs-2*. Moreover, we used luciferase reporter assays to show that HLH-11 and its mammalian homolog TFAP4 directly repress transcription driven by the *lipl-3* or *acs-2* promoter (Fig. 6c).

Reviewer #3 (Remarks to the Author):

The key message of the manuscript by Yi Li and colleagues is that starvation results in degradation of HLH-11, a transcription factor. This then brings about a transcriptional change that ultimately promotes fat utilization through enhanced beta oxidation. Overall, the paper is well written and the data are presented in a logical order. There are a number of attractive features to the manuscript and many of the experiments are well performed. However, the conclusions drawn by the author fail to take into consideration a fundamental aspect of lipid metabolism in hermaphroditic *C. elegans* (see below for detail). It is worth noting that lack of consideration of this fundamental issue is applicable to the vast majority of papers pertaining to lipid analysis in *C. elegans*.

We are pleased that the reviewer found our manuscript interesting, and we appreciate his/her comment that our experiments are technically sound and support the conclusions.

We also appreciate the reviewer's insightful suggestion. Based on the results discussed below, we believe that the phenotype of *hlh-11* inactivation is not due to the transfer of lipids from the mother to embryos. We have provided the relevant data and discussion in the revised manuscript.

Major comments:

1) While it is certainly true that *C. elegans* initiate fat breakdown for energetic purposes upon starvation, it is equally true that upon starvation gravid *C. elegans* hermaphrodites massively transfer lipids to their developing progeny in the form of yolk. Yolk is made in the intestine, the very same tissue that is the focus of the investigation in this manuscript. None of the fat measurement techniques used in this manuscript specifically discriminate among lipids that are in storage compartments from those that are in yolk. Since yolk formation requires building of lipoprotein particles, it would be expected that a process such as starvation that promotes build-up and transfer of yolk into progeny would be accompanied by major transcriptional changes in lipid metabolic genes. While burning of lipids to generate energy and remodeling of lipids to make and transfer yolk both involve lipid metabolism genes, they have completely different physiological meanings and cannot be lumped together. The authors should, therefore, provide evidence to distinguish whether the effect seen upon *hlh-11* inactivation pertain to lipid utilization

for energetic demands of the mother (as they suggest) or to transfer of lipids from the mother to embryos which then exit the mother as eggs laid. For example, what happens to lipid content in males or in larval animals deficient in *hlh-11*?

We would like to thank the reviewer for raising this important issue and we appreciate the suggestion. Based on the results listed below, we believe that *hlh-11* inactivation initiates fat breakdown to support the energetic demands of the mothers, but not to transfer lipids to embryos.

- 1) We have followed the reviewer's advice to measure lipid content in males or larval animals deficient in *hlh-11*. We found that males or L4 worms deficient in *hlh-11* also have reduced lipid content (Supplementary Fig. 4c-f).
- 2) Our RNA-seq results showed that knockout of *hlh-11* suppresses the transcription of all six Vitellogenin genes (Supplementary Fig. 4g). Therefore, vitellogenesis, the process of yolk formation, is unlikely to be activated under HLH-11 deficiency.
- 3) In addition to the induction of lipase genes, fatty acid beta-oxidation pathway genes (e.g. ACSs and ACDHs) are also significantly induced in *hlh-11* KO worms (Fig. 3d). This suggests that fatty acids are further broken down through beta-oxidation to provide energy for mothers.
- 4) ORO staining indicated that lipid levels were not elevated in embryos of *hlh-11* knockout animals (Supplementary Fig. 4h).

2) Figure 5 (effects of starvation on stability of the HLH-11 protein): The data, as shown, are convincing. The issue is that the experiments in Figure 5 utilize L4 (e.g. non-gravid) animals while much of the rest of the data (RNAseq, lipid analysis) are conducted on gravid adults. This may seem like a small thing but once again given the enormous effects of the onset of reproduction on lipid metabolism in adult hermaphrodites, these developmental states cannot be considered as equivalent. As suggested above, at least the fat measurements should be done in L4 (ensuring that the animals have not yet committed to generation of embryos).

We have followed the reviewer's suggestion to measure fat content at L4 stage. We found that L4 worms deficient in *hlh-11* also have reduced lipid content (Supplementary Fig. 4e, f).

Other comments:

1) The authors state: "During lipid catabolism, after breakdown of body fats by lysosomal lipases to generates free fatty acids (FFAs), the carbon-carbon bonds of FFAs are further broken down through beta oxidation in mitochondria". In mammals, fatty acids liberated from lipid droplets are directly fed into peroxisomal and mitochondrial oxidation pathways. Lysosome play a role in fats that are imported into cells through lipoproteins. If the intestine is the major site of fat storage, then why are the lipids going through lysosomes prior to being sent to beta oxidations pathways? Can you please clarify?

Lipids stored in lipid droplets can be broken down through both cytosolic lipolysis and lysosome-mediated lipophagy, generating fatty acids for beta-oxidation. This has been shown by several labs (Eyleen J. O'Rourke et al., Nature Cell Biology, 2013; Liu K, Cell Death & Differentiation, 2013; Katharina Papsdorf, Trends Cell Biol, 2018). In addition, one study in mammals indicates that lipid droplet size may dictate the mechanistic process for lipid catabolism (Micah B. Schott et al., J Cell Biol. 2019). There is also a review paper on the roles of cytosolic lipolysis and lipophagy for breaking down fats stored in lipid droplets (Rudolf Zechner et al., Nat. Rev. Mol Cell Biol. 2017).

2) Back to the yolk issue: of course, *acs* and *acdh* genes can play roles in fat oxidation, but they

can also play roles in virtually every other aspect of lipid metabolism, for example, yolk formation. Thus, in the absence of direct evidence, the speculations regarding the roles of these genes should be toned down and the appropriate caveats explicitly stated.

As we discussed above, we have followed the reviewer's advice to measure lipid content in males or larval animals deficient in *hlh-11*. We found that males or L4 worms deficient in *hlh-11* also have reduced lipid content (Supplementary Fig. 4c-f). In addition, our RNA-seq results showed that knockout of *hlh-11* suppresses the transcription of all six Vitellogenin genes (Supplementary Fig. 4g). Therefore, vitellogenesis, the process of yolk formation, is unlikely to be activated under HLH-11 deficiency. Moreover, ORO staining indicated that lipid levels were not elevated in embryos of *hlh-11* knockout animals (Supplementary Fig. 4h). We thank the reviewer for raising this important question. We have followed the reviewer's suggestion to carry out additional experiments and discuss the yolk issue (page 11).

3) While it is perfectly reasonable to bring up the potential interplay of lipid metabolic and innate immunity genes, without directly testing the effects of *hlh-11* on innate immunity, the interpretations should be presented with the appropriate caveats.

We agree with the reviewer. We have toned down our statements and acknowledged the appropriate caveats in the revised manuscript (page 10).

4) *Glo-4* also affects gut granules, which are lysosome related organelles. Thus, it cannot be concluded that the effects seen in *glo-4* mutants are necessarily due to lysosomes.

We thank the reviewer for raising this point. We have performed this experiment again using a lysosome acidity neutralizer, NH_4Cl . We observed that only when we impaired both lysosomes and proteasomes by treating *C. elegans* with NH_4Cl and the proteasome inhibitor Bortezomib, or by treating *glo-4(ok623)* mutants with Bortezomib, we were able to suppress the degradation of HLH-11 proteins upon fasting.

REVIEWERS' COMMENTS

Reviewer #1 (Remarks to the Author):

The authors addressed my concerns.

Reviewer #2 (Remarks to the Author):

The others have that than adequate to help addressing the concerns of this reviewer. The only exception is that some of the new day that raises some questions.

1-why do the authors think that mammalian hlh-11 would behave similarly is a worm system even if it is the equivalent.

2-chip PCR experiments need more explanation. is the expression of the gene depended on Hlh-11?

These two questions should be addressed prior to publication.

Reviewer #3 (Remarks to the Author):

The authors have nicely addressed all of my concerns. Congratulations on a producing an interesting manuscript.

Responses to reviewers:

Reviewer #1 (Remarks to the Author):

The authors addressed my concerns.

We are very pleased that the reviewer is satisfied with our responses.

Reviewer #2 (Remarks to the Author):

The others have that than adequate to help addressing the concerns of this reviewer. The only exception is that some of the new day that raises some questions.

1-why do the authors think that mammalian hlh-11 would behave similarly is a worm system even if it is the equivalent.

Because metabolic regulation is highly conserved across species. In most cases, mammalian genes function through similar molecular mechanisms as their *C. elegans* homologues. It is very common to rescue the phenotype of a mutant worm deficient in a gene of interest with its mammalian homolog, to show that these two genes are indeed conserved.

Please refer to the following references:

1. A Cold-Sensing Receptor Encoded by a Glutamate Receptor Gene, *Cell*, 2019
2. KLHL22 activates amino-acid-dependent mTORC1 signaling to promote tumorigenesis and ageing
3. Human GABARAP can restore autophagosome biogenesis in a *C. elegans* lgg-1 mutant, *Autophagy*, 2014
4. Polymodal Sensory Function of the *Caenorhabditis elegans* OCR-2 Channel Arises from Distinct Intrinsic Determinants within the Protein and Is Selectively Conserved in Mammalian TRPV Proteins, *J Neurosci*, 2005

2-chip PCR experiments need more explanation. is the expression of the gene depended on Hlh-11?

These two questions should be addressed prior to publication.

Yes, the expression of *lip1-3* and *acs-2* are dependent on HLH-11. This has already been shown in our manuscript (Fig. 1f-h and Supplementary Fig. 1a, b). We have followed the reviewer's suggestion to add one sentence to describe the rationale of the ChIP-PCR experiment in page 6.

Reviewer #3 (Remarks to the Author):

The authors have nicely addressed all of my concerns. Congratulations on a producing an interesting manuscript.

We thank the reviewer for the encouraging remarks on our work.